# Geometric Resampling in Nearly Linear Time for Follow-the-Perturbed-Leader with Best-of-Both-Worlds Guarantee in Bandit Problems

**Botao Chen** [* 1]  **Jongyeong Lee** [* 2]  **Junya Honda** [1 3]

## Abstract

This paper studies the complexity and optimality of Follow-the-Perturbed-Leader (FTPL) policy in the $K$-armed bandit problems. FTPL is a promising policy that achieves the Best-of-Both-Worlds (BOBW) guarantee without solving an optimization problem unlike Follow-the-Regularized-Leader (FTRL). However, FTPL needs a procedure called geometric resampling to estimate the loss, which needs $O(K^2)$ per-round average complexity, usually worse than that of FTRL. To address this issue, we propose a novel technique, which we call Conditional Geometric Resampling (CGR), for unbiased loss estimation applicable to general perturbation distributions. CGR reduces the average complexity to $O(K \log K)$ without sacrificing the regret bounds. We also propose a biased version of CGR that can control the worst-case complexity while keeping the BOBW guarantee for a certain perturbation distribution. We confirm through experiments that CGR does not only significantly improve the average and worst-case runtime but also achieve better regret thanks to the stable loss estimation.

## 1. Introduction

The multi-armed bandit (MAB) is a sequential decision-making problem under uncertainty, which describes a model of a gambler playing slot machines. It has been widely applied in practical scenarios, such as online advertising, clinical trials, recommendation systems. In this problem, the player chooses an arm $I_t$ out of $K$ arms in each round $t \in [T] = \{1, 2, \ldots, T\}$ for time horizon $T$. The loss vector $\ell_t = (\ell_{t,1}, \ell_{t,2}, \ldots, \ell_{t,K})^\top \in [0, 1]^K$ is determined by the environment and the player can only observe the incurred loss $\ell_{t,I_t}$ from the chosen arm. The player's objective is to minimize his/her cumulative losses over all rounds. The performance of the player is measured by the pseudo-regret, which describes the gap between the cumulative loss of the player and of the optimal arm fixed in hindsight.

Stochastic setting (Lai & Robbins, 1985; Auer et al., 2002a) and adversarial setting (Auer et al., 2002b; Audibert & Bubeck, 2009) are two fundamental formulations of the environment to determine loss vectors. In the stochastic setting, the losses $\ell_{t,i}$ from arm $i$ are i.i.d. from an unknown but fixed distribution over $[0, 1]$ with expectation $\mu_i$. The suboptimality gap is expressed by $\Delta_i = \mu_i - \mu_{i^*}$ for the optimal arm $i^* \in \arg\min_{i \in [K]} \mu_i$. The optimal (problem-dependent) regret bound given these gaps is known to be $\sum_{i:\Delta_i > 0} O(\frac{\log T}{\Delta_i})$ (Lai & Robbins, 1985), which can be achieved by several policies such as UCB (Auer et al., 2002a; Cappé et al., 2013) and Thompson sampling (Kaufmann et al., 2012; Agrawal & Goyal, 2013; Riou & Honda, 2020), some of which further improve the regret bound by incorporating the dependence on the loss distribution itself.

In the adversarial setting, the loss vectors $\ell_t$ are not assumed to follow any specific distribution, and the environment may choose them based on the history of the decisions. For this setting, the optimal regret bound $O(\sqrt{KT})$ (Auer et al., 2002b) is achieved by some policies, including a family of policies called the Follow-the-Regularized-Leader (FTRL) with appropriate regularization functions (Audibert & Bubeck, 2009; Zimmert & Lattimore, 2019).

In reality, the environment to determine the loss is often unknown, and thus it becomes necessary to develop a policy that is optimal in both stochastic and adversarial settings at the same time. The Tsallis-INF policy (Zimmert & Seldin, 2021) is an FTRL-based policy that can overcome this difficulty. Tsallis-INF achieves $O(\sqrt{KT})$ regret in the adversarial setting and $\sum_{i \neq i^*} O(\frac{\log T}{\Delta_i})$ regret in the stochastic setting, which is called a Best-of-Both-Worlds (BOBW, Bubeck & Slivkins, 2012) guarantee. It has also been shown that FTRL framework can achieve BOBW for the problems beyond the classic bandit problems, such as partial monitoring (Tsuchiya et al., 2023b;c) and combinatorial-semi bandits (Zimmert et al., 2019; Ito, 2021).

---
*Equal contribution [1]Kyoto University, Kyoto, Japan [2]Seoul National University, Seoul, Korea [3]RIKEN AIP, Tokyo, Japan. Correspondence to: Botao Chen <chen.botao.63r@st.kyoto-u.ac.jp>, Jongyeong Lee <jongyeong@snu.ac.kr>.

*Proceedings of the 42$^{nd}$ International Conference on Machine Learning*, Vancouver, Canada. PMLR 267, 2025. Copyright 2025 by the author(s).

One limitation of FTRL is that the arm-selection probability $w_t$ is required to be explicitly computed and stored in each round, which requires to solve an optimization problem involving a regularizer. This becomes problematic in some complex settings, and the framework of Follow-the-Perturbed-Leader (FTPL) is a promising candidate to overcome this limitation. FTPL greedily chooses an arm with the minimum cumulative estimated loss with some random perturbation and does not require to solve an optimization problem nor store the vector $w_t$. These properties have made FTPL effective across various online learning and bandit problems, including combinatorial semi-bandits (Neu & Bartók, 2016), online learning with non-linear losses (Dudík et al., 2020), and MDP bandits (Dai et al., 2022), though the regret guarantees of FTPL are often somewhat weaker than FTRL (see Appendix A for more details of FTPL literature).

From the theoretical viewpoint, FTPL has also been studied for the standard MAB in terms of the duality between FTRL and FTPL, and Kim & Tewari (2019) raised an open problem that FTPL achieving $O(\sqrt{KT})$ regret (if exists) would have perturbations following Fréchet-type distributions. As a solution to this conjecture, it has recently been proved that FTPL with Fréchet perturbation with $\alpha = 2$ achieves the BOBW guarantee (Honda et al., 2023), which was later extended to more general Fréchet-type distributions with some mild conditions (Lee et al., 2024).

From the practical viewpoint, however, the advantage of FTPL for the standard MAB has been somewhat limited compared with the clear computational advantage of FTPL for complex problems. The reason is that FTPL and FTRL generally need an unbiased (or low-bias) estimator for the loss vector $\ell_t$ for each round. In FTRL, the Importance-Weighted (IW) estimator is often used, which is an unbiased estimator for $\ell_t$ using the inverse of the arm-selection probability $w_{t,i}^{-1}$. On the other hand in FTPL, $w_t$ is not explicitly computed and the IW estimator becomes unavailable.

To address this problem, Neu & Bartók (2016) proposed a technique called Geometric Resampling (GR) to obtain an unbiased estimator for $w_{t,i}^{-1}$, but its computational cost is $O(K^2)$ per round in average. On the other hand, $w_t$ for Tsallis-INF (i.e., FTRL with Tsallis-entropy regularizer) can be efficiently computed by, e.g., Newton's method, whose computational cost per iteration is $O(K)$. Though the required number of iterations to keep the theoretical guarantee is formally unknown, it is empirically known that $O(1)$ iterations is sufficient. Therefore, the entire computational cost of Tsallis-INF per round becomes $O(K)$ in practice, which becomes more efficient than FTPL with $O(K^2)$ cost. In addition, very recently Zimmert & Marinov (2024) proposed TS-Prod policy, which is related to the first-order approximation of online mirror descent and Tsallis-INF. TS-Prod provably achieves the BOBW guarantee with a

simple update rule without iterations, though its empirical performance is worse than Tsallis-INF.

**Contribution of the Paper**  This paper proposes a novel technique which we call Conditional Geometric Resampling (CGR), which gives unbiased or low-bias estimators for $w_{t,i}^{-1}$ in FTPL. The idea behind the technique is that, while the original GR "faithfully" simulates the arm-selection procedure of FTPL by resampling the perturbation until termination, CGR resamples the perturbation only from those satisfying a necessary condition for termination. By appropriately choosing this necessary condition we can significantly reduce the required number of resampling $M_t$ while keeping the complexity of resampling once to be $O(K)$.

We propose three variants of CGR (CGR I, II-unbiased, II-biased), all of which achieve the quasi-linear average complexity of $O(K \log K)$ without sacrificing the regret of the original GR with $O(K^2)$ complexity. These variants differ in the detailed guarantees and requirements, which are summarized in Table 1. In summary, while the complexity guarantee of CGR II is stronger than CGR I, CGR II requires that the cumulative distribution $F(x)$ and its inverse are computable. Furthermore, CGR significantly improves the regret from FTPL with GR empirically thanks to the reduced variance of the estimators as we will see in Section 5.

## 2. Problem Setup

In this section, we formulate the problem and introduce the framework of FTPL with geometric resampling. At each round $t \in [T] = \{1, 2, \ldots, T\}$, the environment determines a loss vector $\ell_t = (\ell_{t,1}, \ell_{t,2}, \ldots, \ell_{t,K})^\top \in [0,1]^K$. The player then pulls an arm $I_t \in [K]$, and observes the incurred loss $\ell_{t,I_t}$ associated with the chosen arm.

The loss vector is determined in either a stochastic or adversarial way. In the stochastic setting, loss vectors $\ell_1, \ell_2, \ldots, \ell_T \in [0,1]^K$ are i.i.d. from an unknown but fixed distribution $\mathcal{P}$ over $[0,1]^K$. The expected loss from arm $i$ is denoted by $\mu_i = \mathbb{E}_{\ell \sim \mathcal{P}}[\ell_{t,i}] \in [0,1]$. The suboptimality gap of arm $i$ is $\Delta_i = \mu_i - \mu_{i^*}$, where $i^* \in \arg\min_{i \in [K]} \mu_i$ represents the optimal arm. In this paper, we assume that $i^*$ is unique in the stochastic setting. In adversarial setting, the loss $\ell_t$ is determined adversarially, which may depend on the history of the chosen arm $\{I_s\}_{s=1}^{t-1}$.

We measure the performance of the player in terms of the pseudo-regret $\mathcal{R}(T)$ defined as

$$\mathcal{R}(T) = \mathbb{E}\left[\sum_{t=1}^{T}(\ell_{t,I_t} - \ell_{t,i^*})\right], i^* \in \arg\min_{i \in [K]} \mathbb{E}\left[\sum_{t=1}^{T}\ell_{t,i}\right].$$

### 2.1. Follow-the-Perturbed-Leader

We consider the Follow-the-Perturbed-Leader (FTPL) policy, whose entire procedure is in Algorithm 1. At ev-

*Table 1.* Comparison on the number of resampling $M_t$, computational requirements on the CDF of the perturbation and regret bounds between original geometric resampling (GR) and conditional geometric resampling (CGR). Note that Neu & Bartók (2016) considers combinatorial semi-bandits and the result here is the one when applied to the MAB. The last column is the regret guarantee under appropriate choice of perturbation distribution, which is exponential distribution $\mathcal{E}_1$ with unit scale for Neu & Bartók (2016), Fréchet-type distributions with some conditions in Lee et al. (2024, Theorem 2) for GR, CGR I, CGR II-unbiased, and the Fréchet distribution $\mathcal{F}_2$ with shape 2 for CGR II-biased.

| Algorithms | $\mathbb{E}[M_t]$ | $\mathbb{E}\big[M_t\big|\hat{L}_t\big]$ | $\mathbb{E}\big[M_t\big|\hat{L}_t, I_t\big]$ | Worst-case $M_t$ | Computational Requirement | Regret Bounds |
|---|---|---|---|---|---|---|
| **GR** | $O(K)$ | $O(K)$ | Unbounded | Unbounded | None | BOBW |
| **GR** (Neu & Bartók, 2016) | $O(K)$ | $O(K)$ | $O(\sqrt{KT})$ (under $\mathcal{E}_1$) | $\lceil\sqrt{KT}\rceil$ (under $\mathcal{E}_1$) | None | Adversarial |
| **CGR I** | $O(\log K)$ | $O(\log K)$ | Unbounded | Unbounded | None | BOBW |
| **CGR II-unbiased** | $O(\log K)$ | $O(\log K)$ | $O(K)$ (under $\mathcal{F}_2$) | Unbounded | Explicit $F(x)$ and $F^{-1}(x)$ | BOBW |
| **CGR II-biased** | $O(\log K)$ | $O(\log K)$ | $O(K)$ (under $\mathcal{F}_2$) | $\lceil(K\vee 4)\log t\rceil$ (under $\mathcal{F}_2$) | Explicit $F(x)$ and $F^{-1}(x)$ | BOBW |

---

**Algorithm 1** Follow-the-Perturbed-Leader

1: **Input:** Learning rate $\eta_t$
2: Set $\hat{L}_1 := 0$
3: **for** $t = 1, 2, \ldots, T$ **do**
4:    Sample $r_t = (r_{t,1}, r_{t,2}, \ldots, r_{t,K})$ i.i.d. from $\mathcal{D}$
5:    Pull arm $I_t = \arg\min_{i \in [K]}\{\hat{L}_{t,i} - r_{t,i}/\eta_t\}$ and observe $\ell_{t,I_t}$
6:    Compute an estimator $\widehat{w_{t,I_t}^{-1}}$ for $w_{t,I_t}^{-1}$ by geometric resampling
7:    Set $\hat{\ell}_t := \ell_{t,I_t}\widehat{w_{t,I_t}^{-1}}e_{I_t}$ and $\hat{L}_{t+1} := \hat{L}_t + \hat{\ell}_t$
8: **end for**

---

ery round $t$, FTPL keeps the estimated cumulative loss $\hat{L}_t \in \mathbb{R}^K$ specified later and draws a perturbation vector $r_t = (r_{t,1}, r_{t,2}, \ldots, r_{t,K}) \in \mathbb{R}^K$ from some distribution. Then, it pulls the arm minimizing the perturbed loss $\hat{L}_{t,i} - r_{t,i}/\eta_t$, where $\eta_t$ is the learning rate. Unless specified otherwise, we assume that the components of $r_t = (r_{t,1}, r_{t,2}, \cdots, r_{t,K})$ are i.i.d. from a common distribution $\mathcal{D}$ over $\mathbb{R}$, whose density function and cumulative distribution function (CDF) are denoted respectively by $f(x)$ and $F(x)$. In particular, when $\mathcal{D}$ is supported on $[\nu, \infty)$ for some finite $\nu > -\infty$, we assume $\nu = 0$ without loss of generality by considering the location shift of $\mathcal{D}$.

Given $\hat{L}_t$, the probability of pulling arm $i$ is given by

$$w_{t,i} = \mathbb{P}_{r\sim\mathcal{D}}\left[i = \arg\min_{j \in [K]}\{\hat{L}_{t,j} - r_{t,j}/\eta_t\}\right]$$
$$= \int_{\mathbb{R}} f(z + \eta_t \hat{L}_{t,i})\prod_{j\neq i} F(z + \eta_t \hat{L}_{t,j})\,\mathrm{d}z. \quad (1)$$

When $\mathcal{D}$ is supported over $[0, \infty)$ we can write

$$w_{t,i} = \int_0^\infty f(z + \eta_t \underline{L}_{t,i})\prod_{j\neq i} F(z + \eta_t \underline{L}_{t,j})\,\mathrm{d}z.$$

**Algorithm 2** Geometric Resampling

1: **Input:**
   Chosen arm $I_t$, cumulative loss $\hat{L}_t$, learning rate $\eta_t$
2: Set $m := 0$
3: **repeat**
4:    $m := m + 1$
5:    Sample $r_t' = (r_{t,1}', r_{t,2}', \ldots, r_{t,K}')$ i.i.d. from $\mathcal{D}$
6: **until** $I_t = \arg\min_{i\in[K]}\{\hat{L}_{t,i} - r_{t,i}'/\eta_t\}$
7: Set $\widehat{w_{t,I_t}^{-1}} := m$

---

Here the gap of a vector from its minimum is expressed by underlines, for example, $\underline{L}_t = L_t - \mathbf{1}\min_{i\in[K]} L_{t,i} \in [0, \infty)^K$, where $\mathbf{1}$ is the all-one vector.

### 2.2. Geometric Resampling

Since the loss in every round is partially observable, both FTRL and FTPL generally use an estimator $\hat{\ell}_t$ for loss vector $\ell_t$. The estimator of the cumulative loss $L_t = \sum_{s=1}^{t-1}\ell_s$ is then obtained as $\hat{L}_t = \sum_{s=1}^{t-1}\hat{\ell}_t$.

In many policies for the adversarial setting like FTRL, the Importance-Weighted (IW) estimator $\hat{\ell}_t = (\ell_{t,I_t}/w_{t,I_t})e_{I_t}$ is employed as an unbiased estimator for $\ell_t$, where $e_i$ is a unit vector with the $i$-th component set to one. However, as we can see from (1), $w_{t,I_t}$ is not explicitly computed in FTPL unlike FTRL. To address this limitation, FTPL instead uses an estimator $\widehat{w_{t,i}^{-1}}$ for $w_{t,i}^{-1}$ by the technique called Geometric Resampling (GR) described in Algorithm 2.

In GR employed in Neu & Bartók (2016) and Honda et al. (2023), another perturbation vector $r_t'$ is repeatedly drawn from the same distribution $\mathcal{D}$ until $\arg\min_{i\in[K]}\{\hat{L}_{t,i} - r_{t,i}'/\eta_t\} = I_t$ is satisfied, that is, resampling from $\mathcal{D}$ is repeated until the arm that would be pulled under the perturbation $r_t'$ coincides with the actually pulled arm $I_t$.

Here the chosen arm $I_t$ is determined prior to the initia-

tion of the GR process. Since the stopping condition is satisfied with probability $w_{t,I_t}$ in each iteration, the number of resampling follows geometric distribution with success probability $w_{t,I_t}$, whose expectation is $1/w_{t,I_t}$. Letting $\widehat{w_{t,I_t}^{-1}}$ be the number $m$ of resampling, we observe that $\widehat{w_{t,I_t}^{-1}}$ serves as an unbiased estimator for $w_{t,I_t}^{-1}$.

Denote the number of resampling taken by geometric resampling at round $t$ as $M_t$, which is equal to $\widehat{w_{t,I_t}^{-1}}$ in the original GR explained here. Then, the expectation of $M_t$ given $\hat{L}_t$ is expressed as

$$\mathbb{E}[M_t|\hat{L}_t] = \sum_{i\in[K]} \mathbb{P}[I_t=i|\hat{L}_t]\mathbb{E}[M_t|\hat{L}_t, I_t=i]$$
$$= \sum_{i\in[K]} w_{t,i}\cdot\frac{1}{w_{t,i}} = K.$$

Since generating a perturbation vector $r_t$ from $\mathcal{D}$ needs $K$ random number generations, the expected complexity per round becomes $O(K^2)$. From the viewpoint of the worst-case complexity, it is known that $O(\sqrt{KT\log K})$ regret is achievable even if we terminate the resampling after $O(\sqrt{KT})$ repetitions, which leads to $O(\sqrt{K^3T})$ worst-case complexity (Neu & Bartók, 2016). On the other hand, the per-round complexity of FTRL is usually $O(K)$ if the optimization stops within $O(1)$ iterations. Though FTPL is still efficient for moderate size of $K$ (Honda et al., 2023) thanks to its optimization-free nature, the worse dependence on $K$ motivates us to develop a more efficient version of FTPL.

## 3. Proposed Algorithms

In this section, we introduce a family of algorithms which we call Conditional Geometric Resampling (CGR). This family is designed as an improvement over the original GR, providing an estimator for $w_{t,i}^{-1}$ with better computational efficiency. In the following, we write $\sigma_{t,i}$ to denote the number of arms (including $i$ itself) whose cumulative losses do not exceed $\hat{L}_{t,i}$, that is, $\hat{L}_{t,i}$ is the $\sigma_{t,i}$-th smallest among $\{L_{t,j}\}_{j\in[K]}$ where $\sigma_{t,i}=\sigma_{t,j}$ can happen when $\hat{L}_{t,i}=\hat{L}_{t,j}$. For example, a current best arm $\hat{i}_t^*\in\arg\min_j\hat{L}_{t,j}$ is an arm $i$ such that $\sigma_{t,i}=1$. For notational simplicity, we omit the subscript $t$ in $\sigma_{t,i}$ when the context is clear.

### 3.1. General Idea

Before explaining the specific procedures of the proposed algorithms, we first give the intuition behind them. When we run the original GR, some possible values of $r_t'$ clearly violates the condition for termination. For example, when the pulled arm $I_t$ is not the current best arm $\hat{i}_t^*\in\arg\min_i\hat{L}_{t,i}$, then the resampled perturbation must satisfy $r_{t,I_t}'\geq r_{t,\hat{i}_t^*}'$ so that the condition for termination

$$I_t = \arg\min_{i\in[K]}\{\hat{L}_{t,i}-r_{t,i}'/\eta_t\} \tag{2}$$

is satisfied since $\hat{L}_{t,I_t}\geq\hat{L}_{t,\hat{i}^*}$. Thus it is somewhat "wasteful" to sample $r'$ satisfying $r_{t,I_t}'<r_{t,\hat{i}^*}'$ and it becomes sufficient to resample $r'$ only from those satisfying $r_{t,I_t}'\geq r_{t,\hat{i}^*}'$.

Now let us formalize this idea. Let $\mathcal{A}_t$ be an arbitrary necessary condition for termination given in (2), which may depend on $\hat{L}_t$ and $I_t$. For example, the above discussion corresponds to $\mathcal{A}_t=\{r_{t,I_t}'\geq r_{t,\hat{i}_t^*}'\}$. Then, we can easily derive the following property, whose proof is shown in Appendix B.1.

**Lemma 1.** *Consider resampling of $r_t'$ from $\mathcal{D}$ conditioned on $\mathcal{A}_t$ until (2) is satisfied. Then, the number $M_t$ of resampling satisfies*

$$\mathbb{E}[M_t|\hat{L}_t, I_t] = \frac{\mathbb{P}[\mathcal{A}_t|\hat{L}_t, I_t]}{w_{t,I_t}}.$$

By this lemma we can use $M_t/\mathbb{P}[\mathcal{A}_t|\hat{L}_t, I_t]$ as an unbiased estimator of $w_{t,I_t}^{-1}$. In addition, we can reduce the number of resampling if we take $\mathcal{A}_t$ such that $\mathbb{P}[\mathcal{A}_t|\hat{L}_t, I_t]$ is small.

Here note that we can trivially minimize $\mathbb{P}[\mathcal{A}_t|\hat{L}_t, I_t]$ by taking the termination condition (2) itself as $\mathcal{A}_t$. Still, such a choice makes it difficult to compute $\mathbb{P}[\mathcal{A}_t|\hat{L}_t, I_t]$ and to resample $r_t'$ from the conditional distribution given $\mathcal{A}_t$. In the following algorithms, we construct necessary conditions $\mathcal{A}_t$ such that $\mathbb{P}[\mathcal{A}_t|\hat{L}_t, I_t]$ is small and easily computed, while the resampling from the conditional distribution given $\mathcal{A}_t$ is also efficient.

### 3.2. Conditional Geometric Resampling

Based on the above idea, we propose three variants of conditional geometric resampling called CGR I (Algorithms 3), CGR II-unbiased and CGR II-biased (Algorithms 4). Here, CGR II-unbiased and II-biased consider resampling from the same conditional distribution, and thus we will discuss them together. In the algorithm description, blue parts are the ones different from the original GR.

**CGR I** This version corresponds to the case where we sample $r_t'$ conditioned on $\mathcal{A}_t=\{r_{t,I_t}'=\max_{i:\sigma_i\leq\sigma_{I_t}}r_{t,i}'\}$, that is, the event that $r_{t,I_t}'$ is the largest among the arms whose cumulative estimated losses are no worse than arm $I_t$. By the symmetric nature of the i.i.d. perturbations, sampling from this conditional distribution can be realized by simple operations as shown in Algorithm 3, which only needs the extra value-swapping operation in addition to the original GR. This fact and other properties of CGR I are formalized as follows.

**Lemma 2.** *The sample $r_t'$ obtained by Algorithm 3 follows the conditional distribution of $\mathcal{D}$ given $\mathcal{A}_t=\{r_{t,I_t}'=\max_{i:\sigma_{t,i}\leq\sigma_{t,I_t}}r_{t,i}'\}$. In addition,*

$$\mathbb{P}_{r_t'\sim\mathcal{D}}[\mathcal{A}_t|\hat{L}_t, I_t] = 1/\sigma_{t,I_t}$$

**Algorithm 3** Conditional Geometric Resampling I

1: **Input:** Chosen arm $I_t$, cumulative loss $\hat{L}_t$, learning rate $\eta_t$
2: Set $m := 0$
3: **repeat**
4:    $m := m + 1$
5:    Sample $r'_t = (r'_{t,1}, r'_{t,2}, \ldots, r'_{t,K})$ i.i.d. from $\mathcal{D}$
6:    Swap the values of $r'_{t,i'}$ and $r'_{t,I_t}$, where $i' = \arg\max_{i:\sigma_i \leq \sigma_{I_t}} r'_{t,i}$
7: **until** $I_t = \arg\min_{i \in [K]}\{\hat{L}_{t,i} - r'_{t,i}/\eta_t\}$
8: Set $\widehat{w_{t,I_t}^{-1}} := m\sigma_{I_t}$

---

*and the number $M_t$ of resampling satisfies*

$$\mathbb{E}_{r'_t \sim \mathcal{D}|\mathcal{A}_t}[M_t|\hat{L}_t] \leq \log K + 1.$$

The proof of this lemma is shown in Appendix B.2. Combined with Lemma 1, this result suggests $\widehat{w_{t,I_t}^{-1}} = M_t\sigma_{I_t}$ given by CGR I is an unbiased estimator for $w_{t,I_t}^{-1}$. Moreover, the expected number of resampling per round is bounded by $\log K + 1$, which is independent of $\hat{L}_t$.

**Remark 1.** In CGR I we only use the symmetry of the distribution of $r_t \in \mathbb{R}^K$. As a result, the unbiasedness and Lemma 2 are still valid even if the components of $r_t \in \mathbb{R}^K$ are not independent as far as the joint distribution is symmetric, though the regret bounds discussed in this paper are for independent perturbations.

**CGR II** In this version we assume that $\mathcal{D}$ is supported over $[0, \infty)$. CGR II corresponds to the case where we sample $r'_t$ from $\mathcal{D}$ conditioned on

$$\mathcal{A}_t = \left\{ r'_{t,I_t} = \max_{i:\sigma_i \leq \sigma_{I_t}} r'_{t,i}, \; r'_{t,I_t} \geq \eta_t\hat{\underline{L}}_{t,I_t} \right\}, \quad (3)$$

that is, we impose the extra condition $r_{t,I_t} \geq \eta_t\hat{\underline{L}}_{t,I_t}$ in addition to the condition in CGR I. We also set the maximum number of resampling $G_t \in \mathbb{N} \cup \{\infty\}$, where we refer to the versions $G_t = \infty$ by CGR II-unbiased and $G_t < \infty$ by CGR II-biased. The complete procedure of CGR II is given in Algorithm 4, which is explained below.

Let $\mathcal{D}_m$ be the distribution of the maximum of $m$ i.i.d. samples from $\mathcal{D}$, whose CDF is $F^m(x) = (F(x))^m$. In CGR II we initially sample $r'_{t,I_t}$ conditioned on $\mathcal{A}_t$, which follows $\mathcal{D}_{\sigma_{I_t}}$ truncated over $[\eta_t\hat{\underline{L}}_{t,I_t}, \infty)$ with CDF given by

$$F_{I_t}(x; \eta_t\hat{\underline{L}}_{t,I_t}) = \frac{F^{\sigma_{I_t}}(x) - F^{\sigma_{I_t}}(\eta_t\hat{\underline{L}}_{t,I_t})}{1 - F^{\sigma_{I_t}}(\eta_t\hat{\underline{L}}_{t,I_t})},$$
$$x \geq \eta_t\hat{\underline{L}}_{t,I_t}. \quad (4)$$

**Algorithm 4** Conditional Geometric Resampling II

1: **Input:** Chosen arm $I_t$, cumulative loss $\hat{L}_t$, learning rate $\eta_t$, maximum number of resampling $G_t \in \mathbb{N} \cup \{\infty\}$
2: Set $m := 0$
3: **repeat**
4:    $m := m + 1$
5:    Sample $\{r'_{t,i} \mid \sigma_i > \sigma_{I_t}\}$ i.i.d. from $\mathcal{D}$
6:    Sample $r'_{t,I_t}$ from $\mathcal{D}_{\sigma_{I_t}}$ truncated over $[\eta_t\hat{\underline{L}}_{t,I_t}, \infty)$, whose CDF is (4)
7:    Sample $\{r'_{t,i} \mid \sigma_i \leq \sigma_{I_t}, i \neq I_t\}$ i.i.d. from $\mathcal{D}$ truncated over $[0, r'_{t,I_t}]$, whose CDF is (5)
8: **until** $I_t = \arg\min_{i \in [K]}\{\hat{L}_{t,i} - r'_{t,i}/\eta_t\}$ or $m \geq G_t$
9: Set $\widehat{w_{t,I_t}^{-1}} := m\sigma_{I_t}/\left(1 - F^{\sigma_{I_t}}(\eta_t\hat{\underline{L}}_{t,I_t})\right)$

---

Next, we sample $\{r'_{t,i} \mid \sigma_i \leq \sigma_{I_t}, i \neq I_t\}$ i.i.d. from $\mathcal{D}$ truncated over $[0, r'_{t,I_t})$, whose CDF is given by

$$F_i(x; r'_{t,I_t}) = F(x)/F(r'_{t,I_t}), \qquad x \in [0, r'_{t,I_t}]. \quad (5)$$

Note that the computation of $F(x)$ and $F^{-1}(x)$ is necessary in this algorithm to generate samples from (4) and (5) by the method of inverse transform sampling. We can show that the samples generated by this procedure indeed follow the conditional distribution given $\mathcal{A}_t$.

**Lemma 3.** *In CGR II, $r'_t$ follows the conditional distribution of $\mathcal{D}$ given $\mathcal{A}_t$ in (3). In addition,*

$$\mathbb{P}_{r'_t \sim \mathcal{D}}[\mathcal{A}_t|\hat{L}_t, I_t] = \left(1 - F^{\sigma_{t,I_t}}(\eta_t\hat{\underline{L}}_{t,I_t})\right)/\sigma_{t,I_t}$$

*and the number $M_t$ of resampling satisfies*

$$\mathbb{E}_{r'_t \sim \mathcal{D}|\mathcal{A}_t}[M_t|\hat{L}_t] \leq \log K + 1.$$

*Besides, if $\mathcal{D}$ is the Fréchet distribution with shape 2, then $M_t$ satisfies*

$$\mathbb{E}_{r'_t \sim \mathcal{D}|\mathcal{A}_t}[M_t|\hat{L}_t, I_t] \leq K \vee 4.$$

The proof of this lemma is shown in Appendix B.3. Combined with Lemma 1, we see that

$$\widehat{w_{t,I_t}^{-1}} = M_t\sigma_{I_t}/\left(1 - F^{\sigma_{I_t}}(\eta_t\hat{\underline{L}}_{t,I_t})\right)$$

serves as an unbiased estimator for $w_{t,I_t}^{-1}$ if $G_t = \infty$. In addition, the last statement of this lemma shows that the expected number of resampling is bounded by $K \vee 4$ in the worst case of $\hat{L}_t$ and $I_t$ regardless of the choice of $G_t$ if $\mathcal{D}$ is the Fréchet distribution with shape 2. We will show in Section 4 that the choice $G_t = (K \vee 4)\log t$ is sufficient to guarantee the BOBW regret bound.

### 3.3. Comparison between CGR I and CGR II

Now let us consider the complexity per round, which can be expressed as

$$M_t \times \text{(complexity of resampling once)}.$$

The first factor is analyzed depending on the algorithms and conditionings in Lemmas 2 and 3, which are summarized in Table 1. The second factor is $O(K)$ in both CGR I and II, but its factor usually becomes slightly larger in CGR II due to the computation of $F$ and $F^{-1}$. For this reason, while CGR II always achieves smaller $M_t$ than CGR I, the total computational cost becomes higher in CGR II in most cases as shown in the experiments in Section 5.

## 4. Regret Analysis

In this section, we first show that CGR I and II-unbiased keep the regret bounds for the original GR as a direct consequence of the unbiasedness of the estimators. We then show that CGR II-biased also achieves a regret bound with the same order as the unbiased ones.

### 4.1. Regret Bounds

As we demonstrated in the last section, CGR I and II-unbiased still provide the unbiased estimator for $w_{t,i}^{-1}$, and thus for $\ell_{t,i}$. We can directly apply this property to the BOBW analysis by Honda et al. (2023) and Lee et al. (2024) for Fréchet-type perturbations which we state below for completeness.

**Proposition 4.** *FTPL with CGR I and II-unbiased with learning rate $\eta_t = c/\sqrt{t}$ for $c > 0$ satisfies*

$$\mathcal{R}(T) \leq \begin{cases} O\big(\sqrt{KT}\big) & \text{in adversarial bandits,} \\ O\Big(\sum_{i \neq i^*} \frac{\log T}{\Delta_i}\Big) & \text{in stochastic bandits,} \end{cases}$$

*if the perturbation distribution $\mathcal{D}$ is Fréchet-type and satisfies the conditions in Lee et al. (2024, Theorem 2).*

This proposition is straightforward and we omit the proof. This is because the proofs of Honda et al. (2023) and Lee et al. (2024) only use the variance bound given in (6) below for the property of the unbiased loss estimator as far as the loss estimator $\hat{\ell}_t$ is nonnegative, where the nonnegativity trivially holds for CGR I and II-unbiased.

**Remark 2.** To be more precise, FTPL with CGR I and II-unbiased can attain a slightly better regret guarantee than the one with the original GR. This is because the variance of the estimator $\widehat{w_{t,I_t}^{-1}}$ becomes

$$\text{Var}[\widehat{w_{t,I_t}^{-1}}|\hat{L}_t, I_t]$$
$$= \begin{cases} \frac{1}{w_{t,I_t}^2} - \frac{1}{w_{t,I_t}} & \text{(original GR),} \\ \frac{1}{w_{t,I_t}^2} - \frac{1}{\mathbb{P}(\mathcal{A}_t)w_{t,I_t}} & \text{(CGR I, II-unbiased),} \end{cases} \quad (6)$$

which is smaller in CGR I and II-unbiased. Though the improvement is hidden in the big-O notation, it explains the better empirical performance of CGR given in Section 5.

**Remark 3.** *In particular, CGR II slightly improves the dominant term of the adversarial regret by $0.4\sqrt{KT}$ compared with the existing bound for GR. This fact is formalized in Theorem 10 in Appendix D.*

In CGR II-biased, where the maximum number of resampling $G_t$ is finite, the estimator for $w_{t,i}^{-1}$ becomes biased. Here, we provide the regret analysis of CGR II-biased and show that it can also achieve the BOBW result, as stated in the following theorem.

**Theorem 5.** *FTPL with CGR II-biased with learning rate $\eta_t = c/\sqrt{t}$ for $c > 0$ and maximum number of resampling $G_t = (K \vee 4) \log t$ satisfies that*

$$\mathcal{R}(T) \leq \begin{cases} O\big(\sqrt{KT}\big) & \text{in adversarial bandits,} \\ O\Big(\sum_{i \neq i^*} \frac{\log T}{\Delta_i}\Big) & \text{in stochastic bandits,} \end{cases}$$

*if $\mathcal{D}$ is the Fréchet distribution with shape $2$.*

This result shows that FTPL with CGR II-biased reduces both the worst-case and average complexities while preserving the same order of the regret bound when using the Fréchet perturbation with shape $2$. Here note that the GR by Neu & Bartók (2016) achieves $O(\sqrt{KT \log K})$ regret for the adversarial setting with the maximum number of resampling $G_t = O(\sqrt{KT})$. Therefore our result shows that the BOBW guarantee is achievable by significantly better maximum number of resampling in most cases.

**Remark 4.** Our choice $G_t = O(K \log t)$ is slightly worse than $G_t = O(\sqrt{KT})$ in Neu & Bartók (2016) if $T \approx K$, but it is just because their goal is to bound the bias by $O(\sqrt{KT})$ rather than $O(\log T)$. In fact, by following the same argument as Theorem 5 we can see that $G_t = O(K \log(T/K))$ is enough to achieve $O(\sqrt{KT})$ bias, which is no worse than $G_t = O(\sqrt{KT})$ whenever $K \leq T$.

### 4.2. Proof Sketch for Biased Estimator

In this section we give a sketch of a proof of Theorem 5. The complete proof is given in Appendix C. We begin with the following regret decomposition similar to Neu & Bartók (2016, Lemma 5).

**Lemma 6.** *The expected regret of FTPL with CGR II satisfies*

$$\mathcal{R}(T) \leq \sum_{t=1}^{T} \mathbb{E}\Big[\big\langle \hat{\ell}_t, w_t - e_{i^*} \big\rangle\Big]$$
$$+ \sum_{t=1}^{T} \sum_{i \in [K]} \mathbb{E}\left[ w_{t,i}\bigg(1 - \frac{w_{t,i}}{\mathbb{P}(\mathcal{A}_t|\hat{L}_t, I_t = i)}\bigg)^{G_t} \right].$$

The restriction on the maximum number of resampling always reduces the expected value of $\widehat{w_{t,i}^{-1}} = M_t/\mathbb{P}(\mathcal{A}_t|\cdot)$ compared to the case $G_t = \infty$. By using this fact we can show that the first term on the RHS can be directly bounded by the one with the unbiased estimator in Honda et al. (2023). Therefore, it becomes sufficient to consider the second term, which is bounded as follows.

**Lemma 7.** *When $\mathcal{D}$ is the Fréchet distribution with shape $2$ and $G_t = (K \vee 4) \log t$, FTPL with CGR II-biased satisfies*

$$\sum_{t=1}^{T} \sum_{i \in [K]} \mathbb{E}\left[ w_{t,i} \left( 1 - \frac{w_{t,i}}{\mathbb{P}(\mathcal{A}_t|\hat{L}_t, I_t = i)} \right)^{G_t} \right] \leq \log T.$$

This lemma shows that the additional regret introduced by the bias is at most logarithmic in $T$ independent of $K$.

The key to this proof is bounding $w_{t,i}/\mathbb{P}(\mathcal{A}_t|\cdot)$ from below by a constant independent of $\hat{L}_t$. When we consider the Fréchet perturbation we can derive simple bounds on $w_{t,i}$ and $\mathbb{P}(\mathcal{A}_t|\cdot)$ in terms of $\sigma_{I_t}$ and $\eta_t \underline{\hat{L}}_{t,I_t}$. From these bounds we can show $w_{t,i}/\mathbb{P}(\mathcal{A}_t|\cdot) \geq 1/(K \vee 4)$ by considering the worst case of $\sigma_{I_t}$ and $\eta_t \underline{\hat{L}}_{t,I_t}$. We expect that this result can be generalized to more general Fréchet-type distributions by using bounds on $w_t$ for various cases in Lee et al. (2024). Still, it requires heavy case-by-case analysis and we leave it to future work.

## 5. Experiments

In this section we give results of experiments on the number of resampling of geometric resampling and compare the regret and computational efficiency with existing policies. All of the figures are the results of 100 trials, with shaded areas representing confidence intervals computed using standard deviations. The experiments were conducted on an AMD EPYC 7763 CPU, implemented in Python 3.9 using the NumPy library. The code is available at: https://github.com/BotaoChen123/FTPL-CGR[1].

**Environments** Following Zimmert & Seldin (2021), we consider the stochastically constrained adversarial setting and the stochastic setting. Since we observed the same tendency between them, we only give the results for the former setting here and the results for the stochastic setting is given in the appendix. The details of the instances as well as the additional results are given in Appendix E.

**Policies** We compare FTPL with three variants of conditional geometric resampling (CGR I, CGR II-unbiased, CGR II-biased) with FTPL with geometric resampling, which we will respectively write FTPL CGR I,

FTPL CGR II-U, FTPL CGR II-B, and FTPL GR. In all the experiments we used Fréchet distribution with shape $\alpha = 2$ for the perturbation of FTPL.

We also compare the performance with Tsallis-INF and TS-Prod. Tsallis-INF is FTRL with Tsallis-entropy regularizer. In our experiments, $w_t$ in this policy was computed by Newton's method. For the loss estimator of Tsallis-INF, we consider Importance-Weighted (IW) estimator and Reduced-Variance estimator (RV, Zimmert & Seldin, 2021), the latter of which has better regret bounds than the former. We denote Tsallis-INF with these estimators respectively as T-INF IW and T-INF RV.

We used the same learning rate for T-INF as that of Zimmert & Seldin (2021) ($\eta_t = c/\sqrt{t}$ with $c = 2$ for IW and $c = 4$ for RV). Since FTPL in this paper and those in Lee et al. (2024); Honda et al. (2023) are designed to mimic T-INF IW, we use the same learning rate for FTPL as that of T-INF IW.

TS-Prod is a variant of Prod (Cesa-Bianchi et al., 2007), serving as an approximation of T-INF. Unlike T-INF, TS-Prod requires only a one-step update rather than solving an optimization problem, resulting in more efficient computation with a complexity of $O(K)$. We used the same learning rate for TS-Prod as that of Zimmert & Marinov (2024).

**Number of Resampling** Figure 1 shows the number of resampling at each round for adversarial setting on a log scale. From this figure we see that the number of resampling is significantly and stably kept small under all the variants of FTPL CGR. In particular, the medians of FTPL CGR II-U and FTPL CGR II-B are one, that is, resampling terminates within only one trial for more than half of the rounds.

Furthermore, as observed from the outliers, the number of resampling of FTPL GR is somewhat unstable and sometimes prohibitively large (recall that the plot is in log scale), while FTPL CGR effectively controls the maximum number of resampling. In addition, the actual maximum number of resampling under FTPL CGR II-B was much smaller than the theoretical guarantee of $G_t = (K \vee 4) \log t$, which is for example $G_t \approx 295$ for $K = 32$ and $t = 10000$. For this reason the behaviors of FTPL CGR II-U and FTPL CGR II-B were empirically indistinguishable. To avoid redundancy, we present only the results for FTPL CGR II-U in the figures of runtime and regret performance, simply referring to it as FTPL CGR II.

**Regret Comparison** Figure 3 is the comparison of the regret of FTPL with those of FTPL GR, T-INF and TS-Prod. As can be seen from this result, the regret of FTPL CGR I is slightly better than FTPL GR. Furthermore, the regret of FTPL CGR II is close to T-INF IW, and even better in some settings, despite the fact that FTPL CGR is also based on the IW estimator. As discussed in Remark 2, this improved regret of FTPL CGR seems to be thanks to the reduced

---

[1] Adapted from the code of Tsuchiya et al. (2023a) available at https://github.com/tsuchhiii/bobw-variance/tree/master.

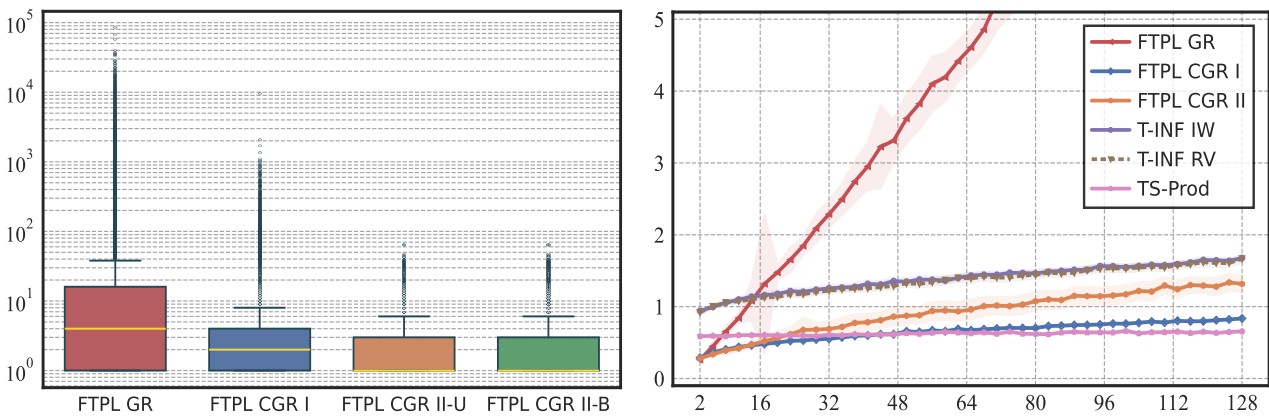

*Figure 1.* Number of resampling for adversarial setting, $K = 32$.

*Figure 2.* Runtime (sec) for adversarial setting and different $K$.

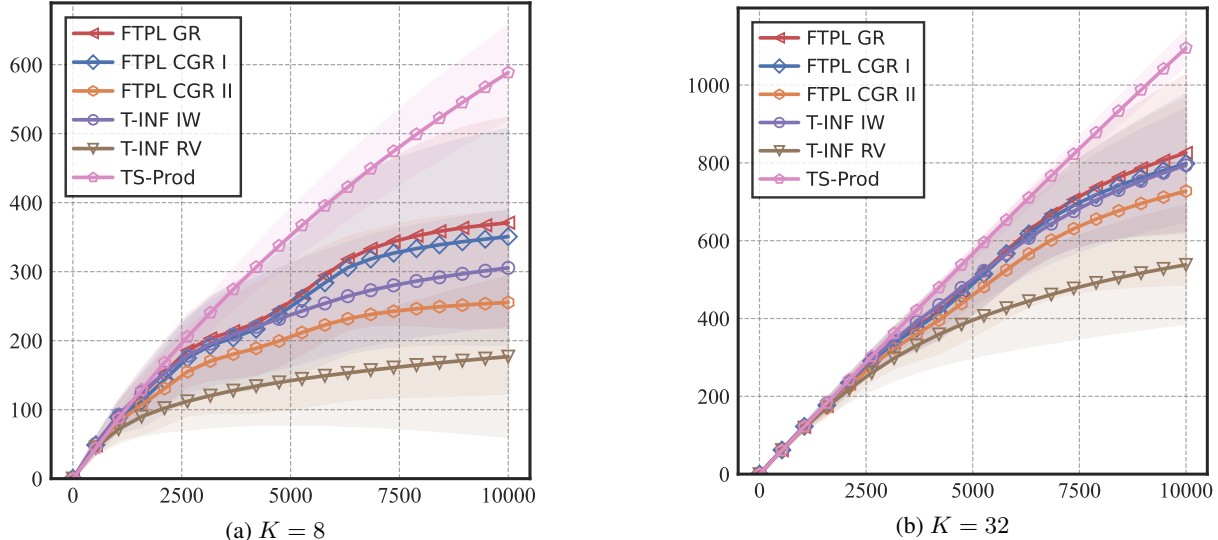

(a) $K = 8$

(b) $K = 32$

*Figure 3.* Pseudo regret in adversarial setting.

variance of the loss estimator. On the other hand, the regret of FTPL CGR II is still worse than T-INF RV. Therefore it is still an important future work to devise a counterpart of RV estimator for FTPL. The regret of TS-Prod is the worst among all compared algorithms, seemingly due to the approximation error introduced by its update rule.

**Computational Efficiency** Figure 2 shows the runtime for arm selection over 10000 rounds of FTPL, T-INF and TS-Prod for varying $K$ from 2 to 128. When $K$ is small enough, both of FTPL GR and FTPL CGR run much faster than T-INF. However, as $K$ increases, the runtime of FTPL GR sharply grows, while those of FTPL CGR, T-INF and TS-Prod are kept small. In particular, the runtime of FTPL CGR and TS-Prod is much smaller than T-INF thanks to its optimization-free nature. Additionally, the runtime of FTPL CGR slightly exceeds TS-Prod when $K$ is large, seemingly due to the additional $\log K$ factor in its complexity.

One interesting observation is that the runtime of FTPL CGR I is consistently better than FTPL CGR II despite its larger number of resampling shown in Figure 1. This comes from the fact that the conditioning procedure of CGR I only needs swapping of the maximum and is very fast.

## 6. Conclusion

In this paper, we proposed Conditional Geometric Resampling (CGR), which is a technique to give unbiased or small-bias estimators for $w_{t,i}^{-1}$. The most important contribution is that, CGR is the first algorithm to enable FTPL to achieve BOBW with average complexity of $O(K \log K)$ without sacrificing the regret. For the second version of CGR with perturbations following Fréchet distribution with shape $\alpha = 2$, we provided a bound of $O(K)$ for the expected number of resampling given the chosen arm, and proved the BOBW property of CGR II-biased with the maximum number of resampling $\lceil (K \vee 4) \log t \rceil$. We empirically demon-

strated that CGR not only significantly improves runtime efficiency, but also improves the regret from FTPL with the original geometric resampling.

## Acknowledgements

BC was supported partially by JST/CREST Innovative Measurement and Analysis (Grant Number JPMJCR2333). JL was supported by the National Research Foundation of Korea (NRF) grant funded by the Korea government (MSIT) (No.RS-2024-00395303). JH was supported partially by JSPS KAKENHI (Grant Number JP25K03184).

## Impact Statement

This paper provides a primarily theoretical contribution to the study of the multi-armed bandit problem, and we believe there are no direct negative ethical or societal consequences that should be highlighted here. Our research may facilitate real-world applications of the FTPL algorithm, such as in recommendation systems and dynamic pricing.

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

# A. Details of FTPL Literature

While originally proposed by Hannan (1957), FTPL gained significant attention for its computational efficiency and simplicity after Kalai & Vempala (2005) introduced it in the context of linear optimization. These properties have made FTPL effective across various online learning and bandit settings, including combinatorial semi-bandits (Neu & Bartók, 2016), online learning with non-linear losses (Dudík et al., 2020), and MDP bandits (Dai et al., 2022). Although variants of FTPL have been developed to improve computational efficiency by avoiding direct optimization, many either sacrifice theoretical performance compared to FTRLs or become computationally expensive when only partial information is available. For example, an earlier efficient FTPL algorithm was shown to suffer $O(T^{2/3})$ regret (Awerbuch & Kleinberg, 2004), while another achieving $O(\sqrt{KT \log K})$ regret required $O(T^2)$ numerical operations per round (Poland, 2005). This inefficiency in bandit settings comes from the nature of FTPL, which avoids explicit optimizations to compute importance weights in closed form. To address this issue, Neu & Bartók (2016) proposed GR in the line of combinatorial semi-bandits, whose reduction to MAB achieves $O(\sqrt{KT \log K})$ regret with $O(K^2)$ numerical operations per round. Further refinements by Honda et al. (2023) and Lee et al. (2024) showed that FTPL with GR can achieve optimal $O(\sqrt{KT})$ regret and even BOBW guarantees for MAB problems when using appropriately chosen perturbation distributions. However, in standard MAB problems, GR often incurs a higher computational cost than FTRL algorithms, despite FTRL requiring the solution of a convex optimization problem each round. This observation contrasts with combinatorial bandit settings, where GR was initially introduced, and motivates the design of a more efficient resampling method for FTPL.

# B. Proofs of Lemmas on Number of Resampling

In this section, we provide the proofs for Lemmas 1, 2 and 3.

For convenience of describing the conditions in the proof, we define an indicator function as

$$\chi_{t,i}(r'_t) = \begin{cases} 1, & \text{if } i = \arg\min_{j \in [K]}\{\hat{L}_t - r'_t/\eta_t\}, \\ 0, & \text{otherwise.} \end{cases}$$

Then, the condition (2) for termination in GR or CGR is equivalent to $\chi_{t,I_t}(r'_t) = 1$, which we simply write as $\chi_t(r'_t) = 1$.

## B.1. Proof of Lemma 1

**Lemma 1 (restated)** *Consider resampling of $r'_t$ from $\mathcal{D}$ conditioned on $\mathcal{A}_t$ until (2) is satisfied. Then, the number $M_t$ of resampling satisfies*

$$\mathbb{E}_{r'_t \sim \mathcal{D}|\mathcal{A}_t}[M_t|\hat{L}_t, I_t] = \frac{\mathbb{P}[\mathcal{A}_t|\hat{L}_t, I_t]}{w_{t,I_t}}.$$

*Proof.* Consider the arm-selection probability $w_{t,I_t}$ with the condition $\mathcal{A}_t$. $w_{t,I_t}$ can be expressed as

$$\begin{aligned} w_{t,I_t} &= \mathbb{P}[\chi_t(r'_t) = 1|\hat{L}_t, I_t] \\ &= \mathbb{P}[\chi_t(r'_t) = 1|\mathcal{A}_t, \hat{L}_t, I_t]\mathbb{P}[\mathcal{A}_t|\hat{L}_t, I_t] + \mathbb{P}[\chi_t(r'_t) = 1|\mathcal{A}_t^c, \hat{L}_t, I_t]\mathbb{P}[\mathcal{A}_t^c|\hat{L}_t, I_t]. \end{aligned} \tag{7}$$

Note that $\mathcal{A}_t$ is an arbitrary necessary condition for $\chi_t(r'_t) = 1$, which implies that

$$\mathbb{P}[\chi_t(r'_t) = 1|\mathcal{A}_t^c, \hat{L}_t, I_t] = 0.$$

Therefore, from (7) we have

$$w_{t,I_t} = \mathbb{P}[\chi_t(r'_t) = 1|\mathcal{A}_t, \hat{L}_t, I_t]\mathbb{P}[\mathcal{A}_t|\hat{L}_t, I_t]. \tag{8}$$

Now we consider the expected number of resampling $M_t$. Recall that $r'_t$ is sampled from $\mathcal{D}$ conditioned on $\mathcal{A}_t$ until (2) is satisfied, that is, $\chi_t(r'_t) = 1$. Then $M_t$ follows geometric distribution with probability mass function

$$\mathbb{P}[M_t = m|\hat{L}_t, I_t] = \left(1 - \mathbb{P}[\chi_t(r'_t) = 1|\mathcal{A}_t, \hat{L}_t, I_t]\right)^{m-1}\mathbb{P}[\chi_t(r'_t) = 1|\mathcal{A}_t, \hat{L}_t, I_t].$$

Therefore, the expected number of resampling given $\hat{L}_t$ and $I_t$ is expressed as

$$
\begin{aligned}
\mathbb{E}_{r'_t \sim \mathcal{D}|\mathcal{A}_t}[M_t|\hat{L}_t, I_t] &= \mathbb{P}[\chi_t(r'_t) = 1|\mathcal{A}_t, \hat{L}_t, I_t] \sum_{n=1}^{\infty} n\Big(1 - \mathbb{P}[\chi_t(r'_t) = 1|\mathcal{A}_t, \hat{L}_t, I_t]\Big)^{n-1} \\
&= \mathbb{P}[\chi_t(r'_t) = 1|\mathcal{A}_t, \hat{L}_t, I_t]/\Big(\mathbb{P}[\chi_t(r'_t) = 1|\mathcal{A}_t, \hat{L}_t, I_t]\Big)^2 \\
&= 1/\mathbb{P}[\chi_t(r'_t) = 1|\mathcal{A}_t, \hat{L}_t, I_t].
\end{aligned} \tag{9}
$$

Combining (8) and (9), we obtain

$$
\mathbb{E}_{r'_t \sim \mathcal{D}|\mathcal{A}_t}[M_t|\hat{L}_t, I_t] = \frac{\mathbb{P}[\mathcal{A}_t|\hat{L}_t, I_t]}{w_{t,I_t}}.
$$

$\square$

## B.2. Proof of Lemma 2

**Lemma 2 (restated)** *The sample $r'_t$ obtained by Algorithm 3 follows the conditional distribution of $\mathcal{D}$ given $\mathcal{A}_t = \{r'_{t,I_t} = \max_{i:\sigma_i \le \sigma_{I_t}} r'_{t,i}\}$. In addition,*

$$
\mathbb{P}_{r'_t \sim \mathcal{D}}[\mathcal{A}_t|\hat{L}_t, I_t] = 1/\sigma_{I_t}
$$

*and the number $M_t$ of resampling satisfies*

$$
\mathbb{E}_{r'_t \sim \mathcal{D}|\mathcal{A}_t}[M_t|\hat{L}_t] \le \log K + 1.
$$

*Proof.* Let $\mathbb{P}^*[\cdot]$ denote the probability distribution of $r'_t$ after the value-swapping operation. We have

$$
\begin{aligned}
&\mathbb{P}^*\left[\bigcap_{i:\sigma_i \le \sigma_{I_t}} \{r'_{t,i} \le a_i\}\,\Big|\,\hat{L}_t, I_t\right] \\
&= \sum_{j:\sigma_j \le \sigma_{I_t}} \mathbb{P}\left[\bigcap_{i:\sigma_i \le \sigma_{I_t}, i \notin \{j, I_t\}} \{r'_{t,i} \le a_i\}, r'_{t,j} \le a_{I_t}, r'_{t,I_t} \le a_j, r'_{t,j} = \max_{i:\sigma_i \le \sigma_{I_t}} r'_{t,i}\,\Big|\,\hat{L}_t, I_t\right] \\
&= \sum_{j:\sigma_j \le \sigma_{I_t}} \mathbb{P}\left[\bigcap_{i:\sigma_i \le \sigma_{I_t}, i \notin \{j, I_t\}} \{r'_{t,i} \le a_i\}, r'_{t,j} \le a_{I_t}, r'_{t,I_t} \le a_j\,\Big|\,r'_{t,j} = \max_{i:\sigma_i \le \sigma_{I_t}} r'_{t,i}, \hat{L}_t, I_t\right] \mathbb{P}\left[r'_{t,j} = \max_{i:\sigma_i \le \sigma_{I_t}} r'_{t,i}\,\Big|\,\hat{L}_t, I_t\right]
\end{aligned} \tag{10}
$$

By the symmetry of $r'_t \in [0, \infty)^K$, we have

$$
\mathbb{P}\left[r'_{t,j} = \max_{i:\sigma_i \le \sigma_{I_t}} r'_{t,i}\,\Big|\,\hat{L}_t, I_t\right] = \mathbb{P}\left[r'_{t,I_t} = \max_{i:\sigma_i \le \sigma_{I_t}} r'_{t,i}\,\Big|\,\hat{L}_t, I_t\right]. \tag{11}
$$

for any $j$ suth that $\sigma_j \le \sigma_{I_t}$. Then we have

$$
\begin{aligned}
1 &= \mathbb{P}\left[\bigcup_{j:\sigma_j \le \sigma_{I_t}} \{r'_{t,j} = \max_{i:\sigma_i \le \sigma_{I_t}} r'_{t,i}\}\,\Big|\,\hat{L}_t, I_t\right] \\
&= \sum_{j:\sigma_j \le \sigma_{I_t}} \mathbb{P}\left[r'_{t,j} = \max_{i:\sigma_i \le \sigma_{I_t}} r'_{t,i}\,\Big|\,\hat{L}_t, I_t\right] \\
&= \sigma_{I_t} \mathbb{P}\left[r'_{t,I_t} = \max_{i:\sigma_i \le \sigma_{I_t}} r'_{t,i}\,\Big|\,\hat{L}_t, I_t\right],
\end{aligned}
$$

which means that (11) is equal to $1/\sigma_{I_t}$. Therefore, from (10) we have

$$\mathbb{P}^*\left[\bigcap_{i:\sigma_i \le \sigma_{I_t}}\{r'_{t,i} \le a_i\}\Big|\hat{L}_t, I_t\right] =$$
$$\frac{1}{\sigma_{I_t}}\sum_{j:\sigma_j \le \sigma_{I_t}}\mathbb{P}\left[\bigcap_{i:\sigma_i \le \sigma_{I_t}, i\notin\{j,I_t\}}\{r'_{t,i} \le a_i\}, r'_{t,j} \le a_{I_t}, r'_{t,I_t} \le a_j\Big|r'_{t,j} = \max_{i:\sigma_i \le \sigma_{I_t}}r'_{t,i}, \hat{L}_t, I_t\right]. \quad (12)$$

By symmetry, each probability term on RHS of (12) is equal. Therefore, we have

$$\mathbb{P}^*\left[\bigcap_{i:\sigma_i \le \sigma_{I_t}}\{r'_{t,i} \le a_i\}\Big|\hat{L}_t, I_t\right] = \mathbb{P}\left[\bigcap_{i:\sigma_i \le \sigma_{I_t}}\{r'_{t,i} \le a_i\}\Big|\mathcal{A}_t, \hat{L}_t, I_t\right],$$

which means that CGR I samples $r'_t$ from the conditional distribution of $\mathcal{D}$ given $\mathcal{A}_t = \{r'_{t,I_t} = \max_{i:\sigma_i \le \sigma_{I_t}} r'_{t,i}\}$. Combining this fact with Lemma 1, for CGR I we have

$$\mathbb{E}_{r'_t \sim \mathcal{D}|\mathcal{A}_t}[M_t|\hat{L}_t, I_t] = \frac{1}{\sigma_{I_t}w_{t,I_t}}.$$

Then, the expected number of resampling given $\hat{L}_t$ in CGR I is bounded by

$$\mathbb{E}_{r'_t \sim \mathcal{D}|\mathcal{A}_t}[M_t|\hat{L}_t] = \sum_{i=1}^{K}\mathbb{P}[I_t = i|\hat{L}_t]\mathbb{E}_{r'_t \sim \mathcal{D}|\mathcal{A}_t}[M_t|\hat{L}_t, I_t = i]$$
$$= \sum_{i=1}^{K}w_{t,i}\cdot\frac{1}{\sigma_i w_{t,i}}$$
$$\le 1 + \int_1^K\frac{1}{x}\,dx$$
$$= \log K + 1.$$

$\square$

## B.3. Proof of Lemma 3

**Lemma 3 (restated)** *In CGR II, $r'_t$ follows the conditional distribution of $\mathcal{D}$ given $\mathcal{A}_t$ in (3). In addition,*

$$\mathbb{P}_{r'_t \sim \mathcal{D}}[\mathcal{A}_t|\hat{L}_t, I_t] = \left(1 - F^{\sigma_{I_t}}(\eta_t\hat{\underline{L}}_{t,I_t})\right)/\sigma_{I_t}$$

*and the number $M_t$ of resampling satisfies*

$$\mathbb{E}_{r'_t \sim \mathcal{D}}[M_t|\hat{L}_t] \le \log K + 1.$$

*Besides, if $\mathcal{D}$ is the Fréchet distribution with shape 2, then $M_t$ satisfies*

$$\mathbb{E}_{r'_t \sim \mathcal{D}|\mathcal{A}_t}[M_t|\hat{L}_t, I_t] \le K \vee 4.$$

*Proof.* Recall that $\mathcal{D}_{\sigma_{I_t}}$ be the distribution of the maximum of $\sigma_{I_t}$ i.i.d. samples from $\mathcal{D}$, whose cumulative distribution function is given by

$$F^{\sigma_{I_t}}(x) = (F(x))^{\sigma_{I_t}}.$$

Then, the probability density function of $\mathcal{D}_{\sigma_{I_t}}$ is expressed as

$$f^{\sigma_{I_t}}(x) = \begin{cases} \sigma_{I_t}f(x)F^{\sigma_{I_t}-1}(x), & x \in [0, \infty), \\ 0, & \text{otherwise.} \end{cases}$$

Since $r'_{t,I_t}$ is sampled from the truncated distribution of $\mathcal{D}_{\sigma_{I_t}}$ with support $\left[\eta_t \hat{\underline{L}}_{t,I_t}, \infty\right)$, it follows the probability density function

$$f_{I_t}(x; \eta_t \hat{\underline{L}}_{t,I_t}) = \begin{cases} \sigma_{I_t} f(x) F^{\sigma_{I_t}-1}(x) / \left(1 - F^{\sigma_{I_t}}\left(\eta_t \hat{\underline{L}}_{t,I_t}\right)\right), & x \in \left[\eta_t \hat{\underline{L}}_{t,I_t}, \infty\right), \\ 0, & \text{otherwise.} \end{cases}$$

For any $i \neq I_t$ that satisfies $\sigma_i \leq \sigma_{I_t}$, given $r'_{t,I_t} = a_{I_t}$, $r'_{t,i}$ is sampled i.i.d. from the truncated distribution of $\mathcal{D}$ with support $[0, a_{I_t}]$. Therefore, it follows the probability density function

$$f_i(x; a_{I_t}) = \begin{cases} f(x)/F(a_{I_t}), & x \in [0, a_{I_t}], \\ 0, & \text{otherwise.} \end{cases}$$

Given $\hat{\underline{L}}_{t,I_t}$, the joint probability density of $r'_{t,i}$ for $i$ satisfying $\sigma_i \leq \sigma_{I_t}$ is denoted by $f^K(\cdot; \eta_t \hat{\underline{L}}_{t,I_t})$. Then, for $a = (a_1, a_2, \ldots, a_K)$ we have

$$\begin{aligned} f^K(a; \eta_t \hat{\underline{L}}_{t,I_t}) &= f_{I_t}(a_{I_t}; \eta_t \hat{\underline{L}}_{t,I_t}) \prod_{i:\sigma_i \leq \sigma_{I_t}, i \neq I_t} f_i(a_i; a_{I_t}) \\ &= \begin{cases} \sigma_{I_t} \prod_{i:\sigma_i \leq \sigma_{I_t}} f(a_i) / \left(1 - F^{\sigma_{I_t}}\left(\eta_t \hat{\underline{L}}_{t,I_t}\right)\right), & \text{if } a_{I_t} = \max_{i:\sigma_i \leq \sigma_{I_t}} a_i, a_{I_t} \geq \eta_t \hat{\underline{L}}_{t,I_t}, \\ 0, & \text{otherwise.} \end{cases} \end{aligned}$$

Let $\mathbb{P}^*[\cdot]$ denote the probability distribution of $r'_t$ under the resampling method in CGR II. We have

$$\begin{aligned} \mathbb{P}^*\left[\bigcap_{i:\sigma_i \leq \sigma_{I_t}} \{r'_{t,i} \leq a_i\} \middle| \hat{L}_t, I_t\right] &= \underbrace{\int_0^{a_i} \cdots \int_0^{a_{i'}}}_{\sigma_{I_t} \text{ integrals}} f_{I_t}(z_{I_t}; \eta_t \hat{\underline{L}}_{t,I_t}) \, dz_{I_t} \prod_{i:\sigma_i \leq \sigma_{I_t}, i \neq I_t} f_i(z_i; z_{I_t}) \, dz_i \qquad (13) \\ &= \frac{\sigma_{I_t}}{1 - F^{\sigma_{I_t}}\left(\eta_t \hat{\underline{L}}_{t,I_t}\right)} \int_{\eta_t \hat{\underline{L}}_{t,I_t}}^{a_{I_t}} f(z) \prod_{i:\sigma_i \leq \sigma_{I_t}, i \neq I_t} F(a_i \wedge z) \, dz \\ &= \frac{\sigma_{I_t}}{1 - F^{\sigma_{I_t}}\left(\eta_t \hat{\underline{L}}_{t,I_t}\right)} \mathbb{P}\left[\bigcap_{i:\sigma_i \leq \sigma_{I_t}} \{r'_{t,i} \leq a_i\}, \mathcal{A}_t \middle| \hat{L}_t, I_t\right]. \qquad (14) \end{aligned}$$

Here, $i'$ in (13) is used as a distinct index from $i$, satisfying the constraint $\sigma_{i'} \leq \sigma_{I_t}$. Now we consider the probability of $\mathcal{A}_t = \left\{r'_{t,I_t} = \max_{i:\sigma_i \leq \sigma_{I_t}} r'_{t,i}, r'_{t,I_t} \geq \eta_t \hat{\underline{L}}_{t,I_t}\right\}$ given $\hat{L}_t$ and $I_t$, which is expressed as

$$\begin{aligned} \mathbb{P}[\mathcal{A}_t | \hat{L}_t, I_t] &= \mathbb{P}\left[r'_{t,I_t} = \max_{i:\sigma_i \leq \sigma_{I_t}} r'_{t,i}, r'_{t,I_t} \geq \eta_t \hat{\underline{L}}_{t,I_t} \middle| \hat{L}_t, I_t\right] \\ &= \int_{\eta_t \hat{\underline{L}}_{t,I_t}}^{\infty} f(z) F^{\sigma_{I_t}-1}(z) \, dz \\ &= \int_{\eta_t \hat{\underline{L}}_{t,I_t}}^{\infty} F^{\sigma_{I_t}-1}(z) \, dF(z) \\ &= \left(1 - F^{\sigma_{I_t}}\left(\eta_t \hat{\underline{L}}_{t,I_t}\right)\right) / \sigma_{I_t}. \qquad (15) \end{aligned}$$

Combining (14) and (15), we have

$$\begin{aligned} \mathbb{P}^*\left[\bigcap_{i:\sigma_i \leq \sigma_{I_t}} \{r'_{t,i} \leq a_i\} \middle| \hat{L}_t, I_t\right] &= \frac{\mathbb{P}\left[\bigcap_{i:\sigma_i \leq \sigma_{I_t}} \{r'_{t,i} \leq a_i\}, \mathcal{A}_t \middle| \hat{L}_t, I_t\right]}{\mathbb{P}[\mathcal{A}_t | \hat{L}_t, I_t]} \\ &= \mathbb{P}\left[\bigcap_{i:\sigma_i \leq \sigma_{I_t}} \{r'_{t,i} \leq a_i\} \middle| \mathcal{A}_t, \hat{L}_t, I_t\right], \end{aligned}$$

which means that CGR II samples $r'_t$ from $\mathcal{D}$ conditioned on $\mathcal{A}_t = \left\{r'_{t,I_t} = \max_{i:\sigma_i \leq \sigma_{I_t}} r'_{t,i}, r'_{t,I_t} \geq \eta_t \hat{\underline{L}}_{t,I_t}\right\}$.

The value of $\mathbb{P}[\mathcal{A}_t|\hat{L}_t, I_t]$ is given in (15). According to Lemma 1, for CGR II, $M_t$ satisfies

$$\mathbb{E}_{r'_t \sim \mathcal{D}|\mathcal{A}_t}[M_t|\hat{L}_t, I_t] \leq \frac{1 - F^{\sigma_{I_t}}\left(\eta_t \hat{\underline{L}}_{t, I_t}\right)}{\sigma_{I_t} w_{t, I_t}},$$

where the equality holds if and only if $G_t = \infty$.

Then, the expected number of resampling given $\hat{L}_t$ in CGR II is bounded by

$$
\begin{aligned}
\mathbb{E}_{r'_t \sim \mathcal{D}|\mathcal{A}_t}[M_t|\hat{L}_t] &= \sum_{i=1}^{K} \mathbb{P}[I_t = i|\hat{L}_t] \mathbb{E}_{r'_t \sim \mathcal{D}|\mathcal{A}_t}[M_t|\hat{L}_t, I_t = i] \\
&\leq \sum_{i=1}^{K} w_{t,i} \cdot \frac{1 - F^{\sigma_{I_t}}\left(\eta_t \hat{\underline{L}}_{t, I_t}\right)}{\sigma_i w_{t,i}} \\
&\leq 1 + \int_1^K \frac{1}{x}\, dx \\
&= \log K + 1.
\end{aligned}
$$

According to Lemma 1, we have $\mathbb{E}_{r'_t \sim \mathcal{D}|\mathcal{A}_t}[M_t|\hat{L}_t, I_t] = \mathbb{P}[\mathcal{A}_t|\hat{L}_t, I_t]/w_{t,I_t}$. Therefore, to prove $\mathbb{E}_{r'_t \sim \mathcal{D}|\mathcal{A}_t}[M_t|\hat{L}_t, I_t = i]$ $\leq K \vee 4$, we only need to show $\mathbb{P}[\mathcal{A}_t|\hat{L}_t, I_t = i]/w_{t,i} \leq K \vee 4$. See the proof in Appendix C.2. □

## C. Proofs of regret analysis

In this section, we provide the proofs for Lemmas 6 and 7 on the bias and the regret of CGR II-biased.

### C.1. Proof of Lemma 6

Before presenting the proof of Lemma 6, we first provide an explicit expression for the expectation of the loss estimates generated by CGR II, similar to the approach used in Lemma 4 of Neu & Bartók (2016).

**Lemma 8.** *For all $i \in [K]$ and $t$ such that $w_{t,i} > 0$, the loss estimates of CGR II satisfies that*

$$\mathbb{E}\left[\hat{\ell}_{t,i}\Big|\hat{L}_t\right] = \left(1 - \left(1 - \frac{w_{t,i}}{\mathbb{P}(\mathcal{A}_t|\hat{L}_t, I_t = i)}\right)^{G_t}\right)\ell_{t,i}.$$

*Proof.* In CGR II, the probability of pulling an arm $i$, denoted by $q_{t,i}$, is given as

$$q_{t,i} = \mathbb{P}\left(\chi_{t,i}(r'_{t,i}) = 1\Big|\hat{L}_t\right) = \mathbb{P}\left(\chi_{t,i}(r'_{t,i}) = 1\Big|\hat{L}_t, I_t = i, \mathcal{A}_t\right)\mathbb{P}(\mathcal{A}_t|\hat{L}_t, I_t = i) = \frac{w_{t,i}}{\mathbb{P}(\mathcal{A}_t|\hat{L}_t, I_t = i)},$$

where the last equality follows from (8). Let $M_{t,i}$ denote the number of resampling for an arm $i$ at round $t$. Then, we have

$$
\begin{aligned}
\mathbb{E}_{r'_t \sim \mathcal{D}|\mathcal{A}_t}[M_{t,i}|\hat{L}_t] &= \sum_{k=1}^{\infty} k(1 - q_{t,i})^{k-1} q_{t,i} - \sum_{k=G_t}^{\infty} (k - G_t)(1 - q_{t,i})^{k-1} q_{t,i} \\
&= \sum_{k=1}^{\infty} k(1 - q_{t,i})^{k-1} q_{t,i} - (1 - q_{t,i})^{G_t} \sum_{k=G_t}^{\infty} (k - G_t)(1 - q_{t,i})^{k-G_t-1} q_{t,i} \\
&= (1 - (1 - q_{t,i})^{G_t}) \sum_{k=1}^{\infty} k(1 - q_{t,i})^{k-1} q_{t,i} = \frac{1 - (1 - q_{t,i})^{G_t}}{q_{t,i}}.
\end{aligned}
$$

By definition of $\hat{\ell}_{t,i}$, we have

$$\mathbb{E}\left[\hat{\ell}_{t,i}\Big|\hat{L}_t\right] = q_{t,i}\ell_{t,i}\mathbb{E}\left[M_{t,i}\Big|\hat{L}_t\right] = (1 - (1 - q_{t,i})^{G_{t,i}})\ell_{t,i}. \quad \square$$

**Lemma 6 (restated)** *The expected regret of FTPL with CGR II satisfies*

$$\mathcal{R}(T) \leq \sum_{t=1}^{T} \mathbb{E}\Big[\big\langle \hat{\ell}_t, w_t - e_{i^*} \big\rangle\Big] + \sum_{t=1}^{T} \sum_{i \in [K]} \mathbb{E}\left[ w_{t,i} \left( 1 - \frac{w_{t,i}}{\mathbb{P}(\mathcal{A}_t | \hat{L}_t, I_t = i)} \right)^{G_t} \right].$$

*Proof.* The overall proof is a simple reduction of the results in combinatorial semi-bandits (Neu & Bartók, 2016, Lemma 5) to MAB. By definition, we have for any $i \in [K]$

$$\mathbb{E}\Big[ w_{t,i} \hat{\ell}_{t,i} \Big| \hat{L}_t \Big] = w_{t,i} \mathbb{E}\Big[ \hat{\ell}_{t,i} \Big| \hat{L}_t \Big]$$

$$= w_{t,i} \left( 1 - \left( 1 - \frac{w_{t,i}}{\mathbb{P}(\mathcal{A}_t | \hat{L}_t, I_t = i)} \right)^{G_{t,i}} \right) \ell_{t,i} \qquad \text{(by Lemma 8)}$$

$$= w_{t,i} \ell_{t,i} - w_{t,i} \left( 1 - \frac{w_{t,i}}{\mathbb{P}(\mathcal{A}_t | \hat{L}_t, I_t = i)} \right)^{G_{t,i}} \ell_{t,i}$$

$$\geq w_{t,i} \ell_{t,i} - w_{t,i} \left( 1 - \frac{w_{t,i}}{\mathbb{P}(\mathcal{A}_t | \hat{L}_t, I_t = i)} \right)^{G_{t,i}}. \qquad (16)$$

Then, by definition of $\mathcal{R}(T)$, we have

$$\mathcal{R}(T) = \sum_{t=1}^{T} \mathbb{E}[\langle \ell_t, w_t - e_{i^*} \rangle]$$

$$\leq \sum_{t=1}^{T} \mathbb{E}\Big[ \big\langle \hat{\ell}_t, w_t \big\rangle - \langle \ell_t, e_{i^*} \rangle \Big] + \sum_{t=1}^{T} \sum_{i \in [K]} \mathbb{E}\left[ w_{t,i} \left( 1 - \frac{w_{t,i}}{\mathbb{P}(\mathcal{A}_t | \hat{L}_t, I_t = i)} \right)^{G_{t,i}} \right], \qquad \text{(by (16))}$$

which concludes the proof. $\qquad \square$

### C.2. Proof of Lemma 7

**Lemma 7 (restated)** *When $\mathcal{D}$ is the Fréchet distribution with shape 2 and $G_t = (K \vee 4) \log t$, FTPL with CGR II-biased satisfies*

$$\sum_{t=1}^{T} \sum_{i \in [K]} \mathbb{E}\left[ w_{t,i} \left( 1 - \frac{w_{t,i}}{\mathbb{P}(\mathcal{A}_t | \hat{L}_t, I_t = i)} \right)^{G_t} \right] \leq \log T.$$

*Proof.* For notational simplicity, we denote $\mathbb{P}(\mathcal{A}_t | \hat{L}_t, I_t = i)$ by $\mathbb{P}(\mathcal{A}_{t,i})$ in this proof. Since $(1 - x)^a \leq e^{-ax}$, it suffices to show that for any $t \in \mathbb{N}$ and $i \in [K]$

$$\exp\left( -\frac{w_{t,i}}{\mathbb{P}(\mathcal{A}_{t,i})} G_t \right) \leq \frac{1}{t}.$$

To this end, we address the term $\frac{\mathbb{P}(\mathcal{A}_{t,i})}{w_{t,i}}$ and set an appropriate $G_t$ to establish a tight regret upper bound.

By definition of $\mathcal{A}_{t,i}$, when $\sigma_i$ denotes the number of arms such that $\hat{L}_{t,j} \leq \hat{L}_{t,i}$ it holds that

$$\mathbb{P}(\mathcal{A}_{t,i}) = \int_{\underline{\Delta}_{t,i}}^{\infty} f(z) F^{\sigma_i - 1}(z) \mathrm{d}z.$$

When $\mathcal{D}$ is Fréchet distribution with shape 2, whose density and distribution functions are given as

$$f(x) = \frac{2e^{-\frac{1}{x^2}}}{x^3}, \quad \text{and} \quad F(x) = e^{-\frac{1}{x^2}}, \qquad (17)$$

it holds that

$$\mathbb{P}(\mathcal{A}_{t,i}) = \int_{\underline{\lambda}_{t,i}}^{\infty} \frac{2}{z^3} \exp\left(-\frac{\sigma_i}{z^2}\right) dz$$

$$= \int_{0}^{\frac{\sigma_i}{\underline{\lambda}_{t,i}^2}} \frac{1}{\sigma_i} e^{-t} dt \qquad \text{(by } t = \sigma_i/z^2\text{)}$$

$$= \frac{1 - \exp\left(-\frac{\sigma_i}{\underline{\lambda}_{t,i}^2}\right)}{\sigma_i}. \tag{18}$$

Now, let us consider the lower bound on $w_{t,i}$. When $z > 0$ and $\underline{\lambda}_{t,j} \geq \underline{\lambda}_{t,i}$, i.e., $\sigma_j \geq \sigma_i$ we have

$$\frac{1}{(z + \underline{\lambda}_{t,j})^2} \leq \frac{1}{(z + \underline{\lambda}_{t,i})^2}.$$

On the other hand, when $z \geq \underline{\lambda}_{t,i}$ and $\underline{\lambda}_{t,j} \leq \underline{\lambda}_{t,i}$, we have

$$\frac{1}{(z + \underline{\lambda}_{t,j})^2} < \frac{1}{\left(\frac{z + \underline{\lambda}_{t,i}}{2} + \underline{\lambda}_{t,j}\right)^2} < \frac{4}{(z + \underline{\lambda}_{t,i})^2}.$$

Therefore, we obtain

$$w_{t,i} = \int_{0}^{\infty} \frac{2}{(z + \underline{\lambda}_{t,i})^3} \exp\left(-\sum_{j \in [K]} \frac{1}{(z + \underline{\lambda}_{t,j})^2}\right) dz$$

$$\geq \int_{\underline{\lambda}_{t,i}}^{\infty} \frac{2}{(z + \underline{\lambda}_{t,i})^3} \exp\left(-\sum_{j \in [K]} \frac{1}{(z + \underline{\lambda}_{t,j})^2}\right) dz$$

$$\geq \int_{\underline{\lambda}_{t,i}}^{\infty} \frac{2}{(z + \underline{\lambda}_{t,i})^3} \exp\left(-\frac{4(\sigma_i - 1) + K - \sigma_i + 1}{(z + \underline{\lambda}_{t,i})^2}\right) dz$$

$$= \frac{1 - \exp\left(-\frac{3(\sigma_i - 1) + K}{4\underline{\lambda}_{t,i}^2}\right)}{3(\sigma_i - 1) + K}. \tag{19}$$

Let $h(x) = (1 - e^{-x})/x$. Then, by combining (18) and (19), we obtain

$$\frac{\mathbb{P}(\mathcal{A}_{t,i})}{w_{t,i}} \leq \frac{1 - \exp\left(-\frac{\sigma_i}{\underline{\lambda}_{t,i}^2}\right)}{\sigma_i} \frac{3(\sigma_i - 1) + K}{1 - \exp\left(-\frac{3(\sigma_i - 1) + K}{4\underline{\lambda}_{t,i}^2}\right)}$$

$$\leq \frac{4h(x)}{h(y)}, \tag{20}$$

where $x = \sigma_i/\underline{\lambda}_{t,i}^2$ and $y = \frac{3(\sigma_i - 1) + K}{4\underline{\lambda}_{t,i}^2}$. Except the case $\sigma_i = K = 2$, we can see

$$\frac{y}{x} = \frac{3(\sigma_i - 1) + K}{4\sigma_i} \leq \frac{K}{4}$$

and therefore

$$\frac{\mathbb{P}(\mathcal{A}_{t,i})}{w_{t,i}} \leq \sup_{x,y>0: y \leq Kx/4} \frac{4h(x)}{h(y)}$$

$$= \sup_{x>0} \frac{4h(x)}{h(Kx/4)},$$

since $h(y)$ is decreasing. For $K \leq 4$, it is bounded from above by 4 since $h(x) \leq h(Kx/4)$ by $x \geq Kx/4$. For $K \geq 4$, we have

$$\sup_{x>0} \frac{4h(x)}{h(Kx/4)} = K \sup_{x>0} \frac{1 - e^{-x}}{1 - e^{-Kx/4}} \tag{21}$$

$$= K \sup_{x>0} \left( 1 - \frac{e^{-x} - e^{-Kx/4}}{1 - e^{-Kx/4}} \right) \leq K.$$

Finally, when $\sigma_i = K = 2$, $y/x = 5/8$ holds. Since $h(y)$ is decreasing, we again obtain

$$\frac{\mathbb{P}(\mathcal{A}_{t,i})}{w_{t,i}} \leq \sup_{x>0} \frac{4h(x)}{h(5x/8)} \leq 4.$$

Hence, for all $t \in \mathbb{N}$ and $i \in [K]$, we obtain

$$\frac{\mathbb{P}(\mathcal{A}_{t,i})}{w_{t,i}} \leq K \vee 4.$$

This implies that for any $G_t > 0$

$$\sum_i w_{t,i} \exp\left( -\frac{w_{t,i}}{\mathbb{P}(\mathcal{A}_{t,i})} G_t \right) \leq \sum_i w_{t,i} \exp\left( -\frac{G_t}{K \vee 4} \right)$$

$$= \exp\left( -\frac{G_t}{K \vee 4} \right). \qquad \text{(by } \sum_i w_{t,i} = 1\text{)}$$

Therefore, setting $G_t = (K \vee 4) \log t$ concludes the proof. $\qquad \square$

### C.3. Proof of Theorem 5

**Theorem 5 (restated)** *FTPL with CGR II-biased with learning rate $\eta_t = c/\sqrt{t}$ for $c > 0$ and maximum number of resampling $G_t = (K \vee 4) \log t$ satisfies that*

$$\mathcal{R}(T) \leq \begin{cases} O\left( \sqrt{KT} \right) & \text{in adversarial bandits,} \\ O\left( \sum_{i \neq i^*} \frac{\log T}{\Delta_i} \right) & \text{in stochastic bandits,} \end{cases}$$

*if the perturbation distribution $\mathcal{D}$ is the Fréchet distribution with shape 2.*

*Proof.* The regret of FTPL with CGR II-unbiased with $G_t = (K \vee 4) \log t$ satisfies

$$\mathcal{R}(T) \leq \sum_{t=1}^{T} \mathbb{E}\left[ \left\langle \hat{\ell}_t, w_t - e_{i^*} \right\rangle \right] + \sum_{t=1}^{T} \sum_{i \in [K]} \mathbb{E}\left[ w_{t,i} \left( 1 - \frac{w_{t,i}}{\mathbb{P}(\mathcal{A}_t | \hat{L}_t, I_t = i)} \right)^{G_t} \right] \qquad \text{(by Lemma 6)}$$

$$\leq \sum_{t=1}^{T} \mathbb{E}\left[ \left\langle \hat{\ell}_t, w_t - e_{i^*} \right\rangle \right] + \log T. \qquad \text{(by Lemma 7)}$$

Let $\hat{\ell}_t^{\mathrm{CGR}}$ and $\hat{\ell}_t^{\mathrm{GR}}$ denote the IW estimators of CGR II-unbiased and GR, respectively. By construction, $\mathbb{E}[\hat{\ell}_t^{\mathrm{CGR}}] \leq \mathbb{E}[\hat{\ell}_t^{\mathrm{GR}}]$ holds, allowing the first term on the RHS to be directly bounded by the results for GR in Honda et al. (2023). Further details on adversarial regret can be found in Lemma 9 in Appendix D. For the stochastic setting, as the analysis remains identical to the previous one, we refer readers to Honda et al. (2023, Theorem 2) for further details. $\qquad \square$

## D. Improved regret analysis of CGR II

In this section, we formalize Remark 3 by presenting an improved analysis of the adversarial regret for FTPL with CGR II when the perturbation $\mathcal{D}$ follows the Fréchet distribution with shape parameter 2, as detailed in the following Theorem 10.

To understand how the transition from GR to CGR II affects the regret analysis, we first introduce the necessary notations and summarize relevant prior results. With a slight abuse of notation, we define a function $\phi_i$ for $i \in [K]$ by

$$\phi_i(\lambda) = \int_0^\infty \frac{2}{(z + \underline{\lambda}_i)^3} \exp\left(-\sum_{j=1}^K \frac{1}{(z + \underline{\lambda}_i)^2}\right) \mathrm{d}z.$$

When $\mathcal{D}$ is Fréchet distribution with shape 2, whose density and distribution functions are given in (17), one can see that $\phi_i(\eta_t \hat{\underline{L}}_t) = w_{t,i}$ holds. Let $\phi = (\phi_1, \ldots, \phi_K)$ denote arm-selection probability vector. Then, the following result has been established.

**Lemma 9** (Overall results in Honda et al. (2023)). *In adversarial setting, FTPL with GR, Fréchet distribution with shape* 2, *and learning rate $\eta_t = c/\sqrt{t}$ for $c > 0$ satisfies*

$$\mathcal{R}(T) \leq \underbrace{\sum_{t=1}^T \mathbb{E}\left[\left\langle \hat{\ell}_t, w_{t+1} - w_t \right\rangle\right]}_{stability} + \underbrace{\sum_{t=1}^T \left(\frac{1}{\eta_{t+1}} - \frac{1}{\eta_t}\right) \mathbb{E}[r_{t+1,I_{t+1}} - r_{t+1,i^*}]}_{penalty} + \underbrace{\frac{\sqrt{\pi K}}{c}}_{dependent\ term\ only\ on\ \mathcal{D},\eta_1}$$

$$\leq \sum_{t=1}^T \mathbb{E}\left[\left\langle \hat{\ell}_t, w_{t+1} - w_t \right\rangle\right] + \frac{3.7}{c}\sqrt{KT} + \frac{\sqrt{\pi K}}{c}$$

$$\leq \sum_{t=1}^T \mathbb{E}\left[\left\langle \hat{\ell}_t, \phi(\eta_t \hat{L}_t) - \phi(\eta_t(\hat{L}_t + \hat{\ell}_t)) \right\rangle\right] + \log(T+1) + \frac{3.7}{c}\sqrt{KT} + \frac{\sqrt{\pi K}}{c} \qquad (22)$$

$$\leq \left(12c\sqrt{\pi} + \frac{3.7}{c}\right)\sqrt{KT} + \log(T+1) + \frac{\sqrt{\pi K}}{c},$$

*whose dominant term can be optimized as $\mathcal{R}(T) \leq 17.8\sqrt{KT} + O(\sqrt{K} + \log T)$ when $c = 0.42$.*

In the adversarial setting, the penalty term is bounded by $3.7\sqrt{KT}/c$, independent of the value of the cumulative loss estimator $\hat{L}_t$, as shown through worst-case analysis (see Honda et al., 2023, Lemma 9). In the derivation of (22), by definition of $\phi$, we have

$$w_t - w_{t+1} = \phi(\eta_t \hat{L}_t) - \phi(\eta_{t+1} \hat{L}_{t+1})$$

$$= \phi(\eta_t \hat{L}_t) - \phi(\eta_{t+1}(\hat{L}_t + \hat{\ell}_t))$$

$$= \phi(\eta_t \hat{L}_t) - \phi(\eta_t(\hat{L}_t + \hat{\ell}_t)) + \phi(\eta_t(\hat{L}_t + \hat{\ell}_t)) - \phi(\eta_{t+1}(\hat{L}_t + \hat{\ell}_t)),$$

which implies

$$\sum_{t=1}^T \mathbb{E}\left[\left\langle \hat{\ell}_t, w_t - w_{t+1} \right\rangle\right] = \sum_{t=1}^T \mathbb{E}\left[\left\langle \hat{\ell}_t, \phi(\eta_t \hat{L}_t) - \phi(\eta_t(\hat{L}_t + \hat{\ell}_t)) \right\rangle\right]$$

$$+ \sum_{t=1}^T \mathbb{E}\left[\left\langle \hat{\ell}_t, \phi(\eta_t(\hat{L}_t + \hat{\ell}_t)) - \phi(\eta_{t+1}(\hat{L}_t + \hat{\ell}_t)) \right\rangle\right].$$

Here, the second term is also bounded by $\log(T+1)$, regardless of the use of GR (see Lee et al., 2024, Lemma 8). Therefore, the transition from GR to CGR II affects the upper bound of the stability term, particularly when accounting for the variance of the IW estimator, which is related to the first term in (22). The following theorem, which formalizes Remark 3, shows that using CGR II can further improve the dominant term.

**Theorem 10** (Formal version of Remark 3). *In adversarial setting, FTPL with CGR II, Fréchet distribution with shape* 2, *and learning rate $\eta_t = c/\sqrt{t}$ satisfies*

$$\mathcal{R}(T) \leq \left(11.5c\sqrt{\pi} + \frac{3.7}{c}\right)\sqrt{KT} + \log(T+1) + \frac{\sqrt{\pi K}}{c} + \sum_{t=1}^T \sum_{i \in [K]} \mathbb{E}\left[w_{t,i}\left(1 - \frac{w_{t,i}}{\mathbb{P}(\mathcal{A}_t | \hat{L}_t, I_t = i)}\right)^{G_t}\right],$$

*whose dominant term can be optimized as $\mathcal{R}(T) \leq 17.37\sqrt{KT} + O(\sqrt{K} + \log T)$ when $c = 0.43$ and $G_t \geq (K \vee 4) \log t$.*

Note that the optimized $c$ of CGR II is greater than that of GR, as shown in Lemma 9, implying that FTPL with CGR II-unbiased can further improve the additive constant term.

Before the proof, we define another function introduced in Honda et al. (2023) as

$$I_{i,n}(\lambda) = \int_0^\infty \frac{1}{(z + \lambda_i)^n} \exp\left(-\sum_{j=1}^K \frac{1}{(z + \lambda_j)^2}\right) dz > 0, \tag{23}$$

which satisfies

$$\phi_i(\lambda) = 2I_{i,3}(\underline{\lambda}), \quad \phi_i'(\lambda) = -6I_{i,4}(\underline{\lambda}) + 4I_{i,6}(\underline{\lambda}). \tag{24}$$

Then, the following result was established.

**Lemma 11** (Lemma 5 in Honda et al. (2023)). *If $\lambda_i$ is the $\sigma_i$-th smallest among $\lambda_1, \ldots, \lambda_K$ (ties are broken arbitrarily) then*

$$\frac{I_{i,4}(\underline{\lambda})}{I_{i,3}(\underline{\lambda})} \leq \frac{\sqrt{\pi/\sigma_i}}{2}.$$

*Proof of Theorem 10.* As discussed above, it suffices to show

$$\sum_{t=1}^T \mathbb{E}\left[\left\langle \hat{\ell}_t, \phi(\eta_t \hat{L}_t) - \phi(\eta_t(\hat{L}_t + \hat{\ell}_t))\right\rangle\right] \leq 11.5c\sqrt{\pi KT}.$$

Here, Honda et al. (2023) showed that in their proof of Lemma 7 that

$$\phi_i(\eta_t \hat{L}_t) - \phi_i(\eta_t(\hat{L}_t + (\ell_{t,i}\widehat{w_{t,i}^{-1}})e_i)) = \int_0^{\eta_t \ell_{t,i}\widehat{w_{t,i}^{-1}}} -\phi_i'(\eta_t \hat{L}_t + xe_i)dx$$

$$\leq 6 \int_0^{\eta_t \ell_{t,i}\widehat{w_{t,i}^{-1}}} I_{i,4}(\underline{\eta_t \hat{L}_t + xe_i})dx \qquad \text{(by (24))}$$

$$\leq 6 \int_0^{\eta_t \ell_{t,i}\widehat{w_{t,i}^{-1}}} I_{i,4}(\eta_t \underline{\hat{L}_t})dx \qquad \text{(monotonicity of } I_{i,4})$$

$$\leq 6\eta_t \ell_{t,i} I_{i,4}(\eta_t \underline{\hat{L}_t})\widehat{w_{t,i}^{-1}}.$$

Here, the monotonicity of $I_{i,4}$ in its $i$-th element might not appear trivial. From the definition of $I$ in (23), one can interpret this function as an arm-selection probability of FTPL when $r_i$ is of density of order $(z + \lambda_i)^4$ and $r_j$s for $j \neq i$ follows the Fréchet distribution with shape 2. Therefore, $I_{i,4}$ is monotonically decreasing with respect to the $i$-th argument.

Under CGR II, we obtain

$$\mathbb{E}\left[\widehat{w_{t,I_t}^{-1}}^2 \Big| \hat{L}_t, I_t\right] = \text{Var}\left[\widehat{w_{t,I_t}^{-1}} \Big| \hat{L}_t, I_t\right] + \mathbb{E}^2\left[\widehat{w_{t,I_t}^{-1}} \Big| \hat{L}_t, I_t\right]$$

$$= \frac{2}{w_{t,I_t}^2} - \frac{1}{w_{t,I_t}\mathbb{P}(A_t)}.$$

Recall (20), which shows

$$\frac{\mathbb{P}(A_{t,i})}{w_{t,i}} \leq \frac{4h(x)}{h(y)},$$

where $h(x) = \frac{1-e^{-x}}{x}$, $x = \frac{\sigma_i}{\underline{\lambda}_{t,i}^2}$, and $y = \frac{3(\sigma_i-1)+K}{4\underline{\lambda}_{t,i}^2}$. One can see that for any $i \in [K]$ and $K \geq 2$

$$\frac{y}{x} = \frac{3(\sigma_i - 1) + K}{4\sigma_i} \leq \frac{K}{\sigma_i}.$$

Following the same arguments in (21), we have

$$\frac{\mathbb{P}(A_{t,i})}{w_{t,i}} \leq \frac{4h(x)}{h(y)} \leq \sup_{x>0} \frac{4h(x)}{h(Kx/\sigma_i)} \leq \frac{4K}{\sigma_i},$$

which implies

$$\frac{1}{\mathbb{P}(A_t)} \geq \frac{\sigma_i}{4Kw_{t,i}} \implies \mathbb{E}\left[\widehat{w_{t,I_t}^{-1}}^2 \Big| \hat{L}_t, I_t\right] \leq \left(2 - \frac{\sigma_i}{4K}\right)\frac{1}{w_{t,I_t}^2}.$$

Since $\hat{\ell}_t = (\ell_{t,i}\widehat{w_{t,i}^{-1}})e_i$ when $I_t = i$, we obtain

$$\mathbb{E}\left[\hat{\ell}_{t,i}(\phi_i(\eta_t\hat{L}_t) - \phi_i(\eta_t(\hat{L}_t + \hat{\ell}_t)))\Big|\hat{L}_t\right] \leq \mathbb{E}\left[\chi[I_t = i]\ell_{t,i}\widehat{w_{t,i}^{-1}} \cdot 6\eta_t\ell_{t,i}I_{i,4}(\eta_t\hat{\underline{L}}_t)\widehat{w_{t,i}^{-1}}\Big|\hat{L}_t\right]$$

$$\leq 6\eta_t\mathbb{E}\left[\left(2 - \frac{\sigma_i}{4K}\right)w_{t,i}\frac{\ell_{t,i}^2 I_{i,4}(\eta_t\hat{\underline{L}}_{t,i})}{w_{t,i}^2}\Big|\hat{L}_t\right]$$

$$= 3\eta_t\mathbb{E}\left[\left(2 - \frac{\sigma_i}{4K}\right)\frac{\ell_{t,i}^2 I_{i,4}(\eta_t\hat{\underline{L}}_{t,i})}{I_{i,3}(\eta_t\hat{\underline{L}}_{t,i})}\Big|\hat{L}_t\right] \qquad \text{(by (24))}$$

$$\leq 3\eta_t\mathbb{E}\left[\left(2 - \frac{\sigma_i}{4K}\right)\frac{I_{i,4}(\eta_t\hat{\underline{L}}_{t,i})}{I_{i,3}(\eta_t\hat{\underline{L}}_{t,i})}\Big|\hat{L}_t\right] \qquad (\because \ell_{t,i} \in [0,1])$$

$$\leq \frac{3\sqrt{\pi}}{2}\eta_t\mathbb{E}\left[\left(\frac{2}{\sqrt{\sigma_i}} - \frac{\sqrt{\sigma_i}}{4K}\right)\Big|\hat{L}_t\right]. \qquad \text{(by Lemma 11)}$$

Therefore,

$$\mathbb{E}\left[\left\langle\hat{\ell}_t, \phi(\eta_t\hat{L}_t) - \phi(\eta_t(\hat{L}_t + \hat{\ell}_t))\right\rangle\right] \leq \frac{3\sqrt{\pi}}{2}\eta_t\sum_{i=1}^{K}\left(\frac{2}{\sqrt{\sigma_i}} - \frac{\sqrt{\sigma_i}}{4K}\right)$$

$$\leq 3\sqrt{\pi}\eta_t\left(1 + \int_1^K x^{-1/2}dx\right) - \frac{3\sqrt{\pi}}{2}\eta_t\left(\int_0^K \frac{x^{1/2}}{4K}dx\right)$$

$$= 3\sqrt{\pi}\eta_t(2\sqrt{K} - 1) - \sqrt{\pi}\eta_t\frac{\sqrt{K}}{4}$$

$$\leq \frac{23\sqrt{\pi}}{4}\sqrt{K}\eta_t.$$

By taking summation with $\eta_t = c/\sqrt{t}$, we have

$$\sum_t \mathbb{E}\left[\left\langle\hat{\ell}_t, \phi(\eta_t\hat{L}_t) - \phi(\eta_t(\hat{L}_t + \hat{\ell}_t))\right\rangle\right] \leq \frac{23\sqrt{\pi}c}{4}\sqrt{K}\sum_{t=1}^{T}\frac{1}{\sqrt{t}}$$

$$\leq \frac{23\sqrt{\pi}c}{2}\sqrt{KT},$$

which concludes the proof. □

## E. Details and Additional Results of Experiments

In this appendix, we firstly describe the details of stochastic setting and stochastically constrained adversarial setting (or adversarial setting in short), and then provide results of additional experiments to complement those presented in Section 5.

### E.1. Details of Settings

In our experiments for both settings, the losses of each arms follow the Bernoulli distributions, and we exclusively consider the case where the suboptimality gap $\Delta = 0.125$. For the stochastic setting, which is implemented for the experiments shown in Figures 4, 5, 6, 8c and 8d, the mean losses are decided as $(1 - \Delta)/2$ for the single optimal arm and $(1 + \Delta)/2$ for the other suboptimal arms.

For stochastically constrained adversarial setting, which is implemented for the experiments shown in Figures 1, 2, 3, 8a and 8b, the mean losses for the single optimal arm and the suboptimal arms switch between $(1 - \Delta, 1)$ and $(0, \Delta)$, with the duration of each phase growing exponentially by a factor of 1.6 after every switch. Both settings are similar to those in Zimmert & Seldin (2021).

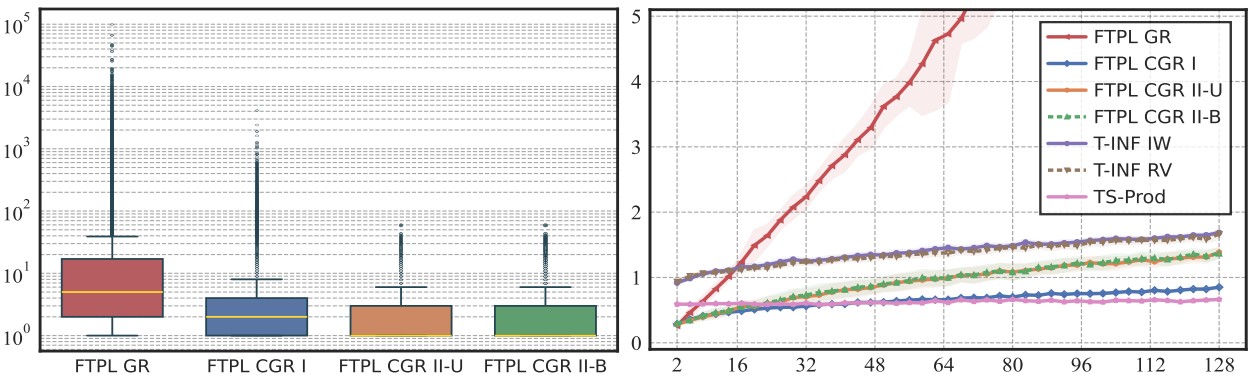

*Figure 4.* Number of resampling for stochastic setting, $K = 32$.   *Figure 5.* Runtime (sec) for stochastic setting and different $K$.

### E.2. Additional Results

Consistently with the experiments in Section 5, the results of 100 trials are shown in this appendix. Figures 4 and 5 show the results of the number of resampling and runtime for stochastic setting. Figures 6 and 7 show the regret performance for stochastic setting and that for adversarial setting, respectively. Note that we include the results both of FTPL CGR II-B and FTPL CGR II-U to explicitly demonstrate their empirical indistinguishability, as discussed in Section 5.

The results for stochastic setting lead to the similar conclusions to those for the adversarial setting, that is, FTPL CGR not only effectively controls the number of resampling and significantly improves the runtime, but also achieves the better empirical regret performance, which seems to be due to the reduced variance of the loss estimator. In addition, Figure 8 provides a visualization of the average and maximum number of resampling at each round over 100 trials for both settings, as a complement to the overview shown in Figures 1 and 4. We can see that, under all the variants of FTPL CGR, the number of resampling at each round is stably controlled at a significantly low value, which is particularly pronounced for FTPL CGR II. Another interesting observation from Figures 8a and 8c is that, for FTPL GR and FTPL CGR I, the average number of resampling fluctuates within a range throughout the horizon, whereas that for FTPL CGR II tends to be gradually reduced as the round progresses, seemingly due to the widening gap of the cumulative estimated losses between the arms.

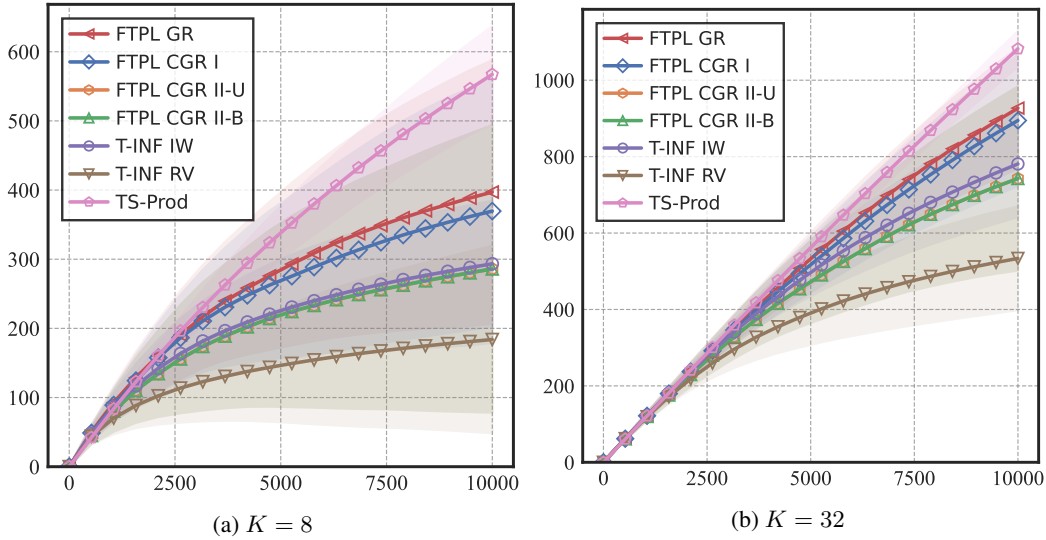

*Figure 6.* Pseudo regret in stochastic setting.

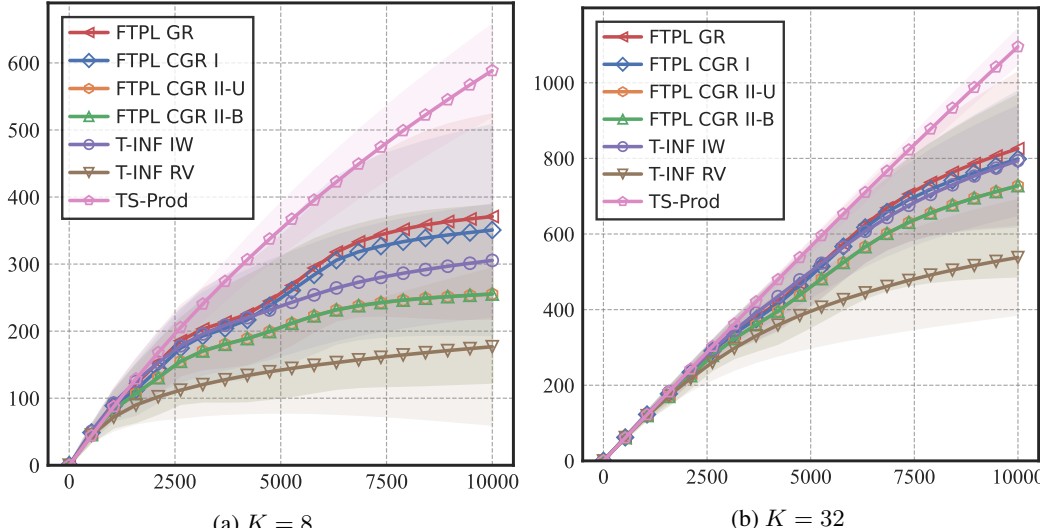

*Figure 7.* Pseudo regret in adversarial setting.

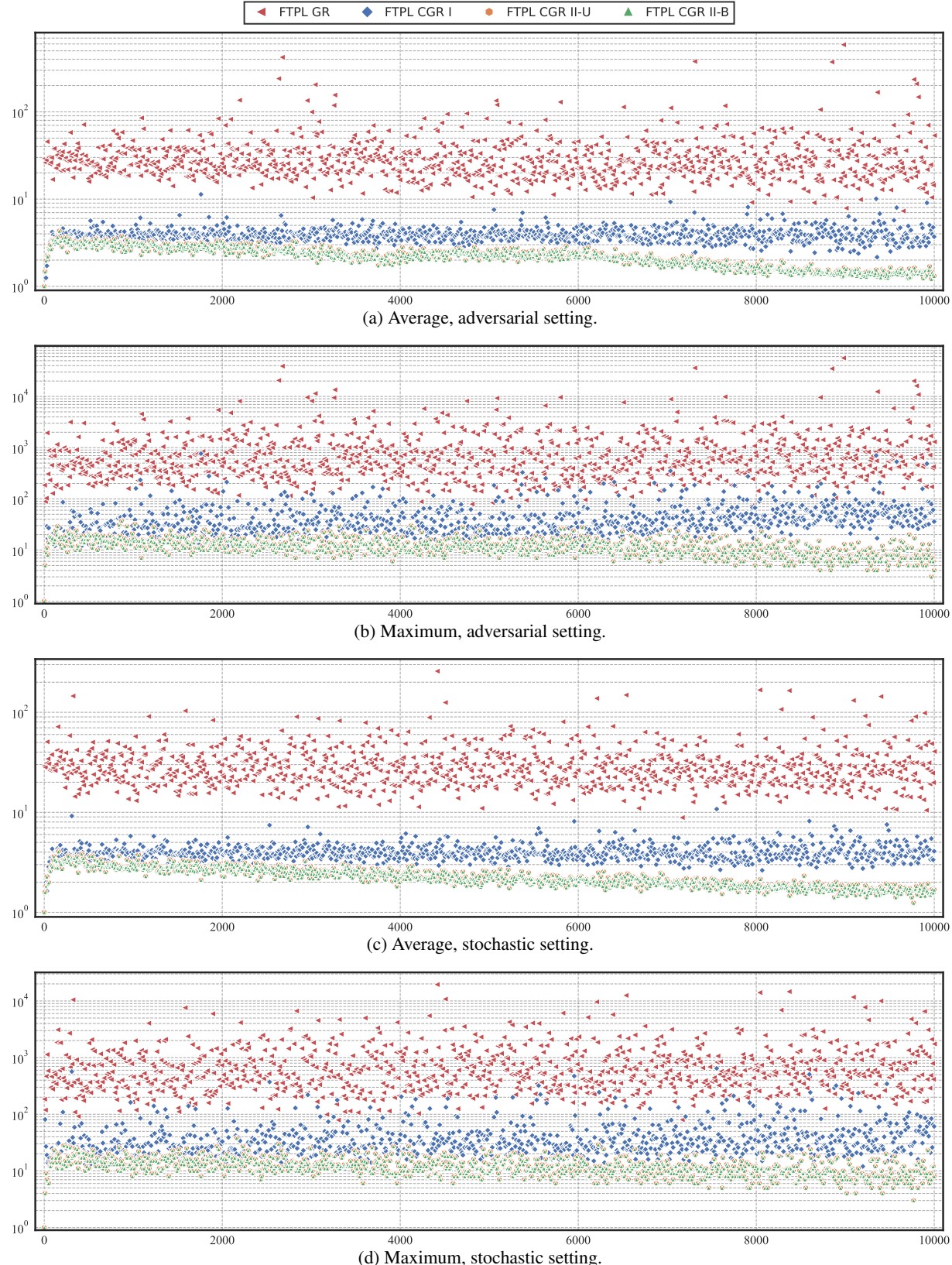

(a) Average, adversarial setting.

(b) Maximum, adversarial setting.

(c) Average, stochastic setting.

(d) Maximum, stochastic setting.

*Figure 8.* Average or Maximum number of resampling at each round for $K = 32$.

