# OpenReview forum: "Geometric Resampling in Nearly Linear Time for Follow-the-Perturbed-Leader with Best-of-Both-Worlds Guarantee in Bandit Problems"
_ICML.cc/2025/Conference — ICML 2025 poster_

### Official Review · Reviewer_tXhY · 2025-03-06

**Overall Recommendation:** 4

**Summary:**

The paper proposed some useful variants of the Geometric Resampling (GR) algorithm called Conditional Geometric Resampling (CGR) and those algorithms improve the sample complexity from O(K^2) to O(Klog(K)) in each round while keeping the BOBW guarantee for a certain perturbation distribution. The experiment result shows the proposed algorithms do perform well in the sample complexity and runtime while keeping a low regret while keeping the BOBW guarantee for a certain perturbation distribution.

## update after rebuttal
I thank the authors for the response. Nothing major has changed, and I keep my score.

**Claims And Evidence:**

Yes, they claim the CGR algorithm has a great improvement compared to the GR algorithm and the experiments do show the improvement for runtime and sampling time.

**Essential References Not Discussed:**

Not to my knowledge.

**Experimental Designs Or Analyses:**

Yes, I checked Section 5.

**Methods And Evaluation Criteria:**

Yes, the paper says the reason that CGR achieves success is that CGR resamples the perturbation only from those satisfying a necessary condition for termination, and that makes sense.

**Other Comments Or Suggestions:**

Could you discuss more about the novelty of the proof of those regret theorems? It is unclear what technical contributions this paper adds.

**Other Strengths And Weaknesses:**

The proposed algorithms greatly improve the sample complexity and runtime from previous GR algorithm.

**Questions For Authors:**

Are there any disadvantages of CGR to the other algorithms (such as Tsallis-INF)?

**Relation To Broader Scientific Literature:**

The paper proposed an experimental beneficial algorithm CGR and its method might have chances to be applied to other time-cost algorithms.

**Theoretical Claims:**

Yes, such as Theorem 6.

---

> ### Author Rebuttal · Authors · 2025-03-30
>
> Thank you for taking the time to review our paper. We have addressed your question and comment below.
>
> **Q1. Are there any disadvantages of CGR to the other algorithms (such as Tsallis-INF)?**
> **A1.**
> Generally, we believe that CGR has no disadvantages over the conventional GR, as it is intuitively just cutting the redundant part of GR, and thus CGR is more effective on each metric.
>
> Although CGR overcomes the computational inefficiency compared with GR, it still inherits the general disadvantages of FTPL. This is mainly the difficulty of the theoretical analysis, such as the difficulty of the analysis for RV estimators and extensions to other settings.
>
> **Q2. Novelty of the regret analysis.**
> **A2.** Thanks for your comment. The main novelty in the regret analysis lies in the analysis of CGR II-B.
> The key difference in the regret analysis stems from the biased estimator, where the relevant term involves both $P(A_t)$ and $w_t$ differently from the original GR.
> By deriving a lower bound on $w_t/P(A_t)$, we obtained the desired result even with a smaller maximum number of resampling.
> While our current analysis is specific to the chosen definition of $A_t$ and the expression of $w_t$, we expect that similar results can be achieved in different settings beyond MAB by appropriately designing $A_t$ and selecting perturbations.

---

### Official Review · Reviewer_tSdm · 2025-03-07

**Overall Recommendation:** 5

**Summary:**

This paper studies FTPL for MAB problem. The authors first propose a general receipe for designing loss estimate procedure. Then they also give several concrete variants under this framework and shows that they enjoy certain improved time complexity while maintain the optimal BOBW guarantee. These theoretical findings are complemented by comprehensive numerical results. The proposed CGR I and II both bring improved empirical time complexity and regret compared to the conventional GR.

**Claims And Evidence:**

Yes

**Essential References Not Discussed:**

n/a

**Experimental Designs Or Analyses:**

Yes

**Methods And Evaluation Criteria:**

Yes

**Other Comments Or Suggestions:**

n/a

**Other Strengths And Weaknesses:**

Strengths

1. Gives a new framework for loss estimate in FTPL.

2. Supporting empirical results.

**Questions For Authors:**

1. any intuitions for why the new proposed scheme would give better empirical regret?

**Relation To Broader Scientific Literature:**

potential audience: people studying bandits/RL/stats

**Theoretical Claims:**

I checked the main body which looks good to me.

---

> ### Author Rebuttal · Authors · 2025-03-30
>
> Thank you for taking the time to review our paper. We have addressed your question below.
>
> **Q1. Any intuitions for why the new proposed scheme would give better empirical regret?**
> **A1.**
> Generally, the variance of the estimator $\widehat{w_{t,I_t}^{-1}}$ introduces an additional regret term in the analysis, which is strongly related to the expected number of resampling in GR and CGR.
> Since the expected number of resampling is significantly reduced in CGR, the variance of $\widehat{w_{t,I_t}^{-1}}$ is also effectively reduced, compared with that of GR (as explained in Remark 2).
> Therefore, the regret from the estimation error of $w_{t,I_t}^{-1}$ gets improved effectively, which also leads to the improved theoretical guarantee in Theorem 5.

---

> > ### Comment · Reviewer_tSdm · 2025-04-07
> >
> > I've read the response from the authors and also the communication between the authors and other reviewers. While this work is limited to MAB, personally I'm still impressed by a new recipe for design and analysis of FTPL, which expends the theoretical understanding on FTPL, so I would still like to support this paper and recommend acceptance.

---

### Official Review · Reviewer_JGty · 2025-03-14

**Overall Recommendation:** 4

**Summary:**

The paper proposes a new estimation procedure based on conditional resampling to improve both theoretically and empirically the sample efficiency of FTPL while maintaining the regret guarantees.

**Claims And Evidence:**

Claims are supported by clear evidence.

**Essential References Not Discussed:**

Prior work discussion seems extensive.

**Experimental Designs Or Analyses:**

I checked all experimental designs and they look valid to me.

**Methods And Evaluation Criteria:**

Evaluation criteria and methods make sense.

**Other Comments Or Suggestions:**

N/A

**Other Strengths And Weaknesses:**

The conditional resampling is a refreshing idea to overcome potential computational issue in FTPL.

**Questions For Authors:**

How would the analysis change in the proposed sampling approach when extending from bandits to semi-bandits?

**Relation To Broader Scientific Literature:**

The paper presents a significant result in FTPL literature.

**Theoretical Claims:**

I have not checked the correctness of proofs.

---

> ### Author Rebuttal · Authors · 2025-03-30
>
> Thank you for taking the time to review our paper. We have addressed your question below.
>
> **Q1. How would the analysis change in the proposed sampling approach when extending from bandits to semi-bandits?**
> **A1.**
> The analysis would not change so much if we are just interested in extending our sampling approach to derive results similar to Neu and Bartók (2016) for semi-bandits.
> Still, their results are only for near-optimal regret bound in the adversarial setting, and extending the BOBW analysis to semi-bandits is significantly difficult.
> In general, the key to the BOBW analysis in FTPL and FTRL is obtaining a uniform bound on $-w_{t,i}'/w_{t,i}^{3/2}$.
> Thus, the expression of $w_{t,i}$ significantly impacts the regret analysis.
> In the semi-bandit setting, the arm selection probability vector $w_t$ no longer lies in the probability simplex since multiple arms can be selected in one round.
> As a result, several techniques used in Honda et al. (2023) and Lee et al. (2024) become no longer applicable.
> In addition, to obtain the desired results for CGR II-B, the events $A_t$ would need to be modified according to the behavior of $w_t$ in semi-bandits.

---

### Official Review · Reviewer_sBDa · 2025-03-14

**Overall Recommendation:** 3

**Summary:**

The paper proposes Conditional Geometric Resampling (CGR) to improve the computational efficiency of the Follow-the-Perturbed-Leader (FTPL) algorithm in the multi-armed bandit problem.
By introducing a carefully selected, necessary stopping condition in the resampling process, CGR reduces the expected computational complexity of standard Geometric Resampling from the quadratic $O(K^2)$ expected per-round time to the near-linear $O(K \\log K)$ while guaranteeing the same best-of-both-worlds regret guarantees from Honda et al. (2023) and Lee et al. (2024).
More precisely, three variants of CGR are introduced: CGR I, CGR II-unbiased, and CGR II-biased.
These variants offer different trade-offs between computational efficiency and estimation bias.
The authors provide both theoretical regret bounds and experimental results that demonstrate lower runtime and improved regret performance compared to previous approaches.

**Claims And Evidence:**

The theoretical claims are clearly stated with complete proofs.
The claims on the empirical performance are also validated by appropriate experiments.

**Essential References Not Discussed:**

The authors seem to have covered the essential related work in their discussion.

**Experimental Designs Or Analyses:**

My main concern about experiments is the omission of FTPL CGR II-B from the experiments other than the number of resampling.
The reason provided by the authors is because the number of resampling is essentially indistinguishable from that of FTPL CGR II-U, and so the other experiments only consider the latter algorithm.
However, it would still be interesting to observe whether the performance is indistinguishable with respect to other performance metrics too (at least for the experiments performed in this work).
For instance, the indistinguishability seems to appear for large enough $K$, e.g. $K \\ge 32$, but the first experiment measuring the cumulative regret considers $K = 8$.

**Methods And Evaluation Criteria:**

Yes.

**Other Comments Or Suggestions:**

- In the Problem Setup section, you define losses $\\ell_t$ in the adversarial setting as being possibly functions of the history of losses and chosen arms, but it seems that you may assume they depend on $(I\_1, \\dots, I\_{t-1})$ w.l.o.g.
- Starting from Section 3, $\\sigma_i$ is introduced but it seems to also depend on the round $t$; making this dependence explicit (e.g., writing $\\sigma\_{i,t}$) is probably clearer.
- Theorem 5 seems more of a remark than an actual theorem.
- In Section 5, it would be clearer to also specify how the confidence intervals around the curves were chosen (e.g., standard deviation?).

**Other Strengths And Weaknesses:**

The main techincal contribution is the design of variants of and the consequent improvement in the computational complexity of FTPL while preserving BOBW guarantees.
Regarding the regret analysis per se, there seems to be no significant difference compared to prior work.
Nevertheless, the time performance is also a relevant factor in practice and the contribution in this direction is interesting.

**Questions For Authors:**

- What is the main technical difficulty in designing a variant of GR that is a sort of counterpart of Tsallis-INF with the Reduced-Variance estimator, especially compared to using importance weighting?
- Did you actually run all the other experiments for FTPL CGR II-B on your own? If so, was there no meaningful difference compared to FTPL CGR II-U?

**Relation To Broader Scientific Literature:**

The adoption of FTPL with geometric resampling to achieve best-of-both-worlds regret guarantees for multi-armed bandits has alredy been considered by previous work.
The main contribution of this paper lies in the design of more clever conditional geometric resampling procedure that employ a carefully chosen necessary stopping condition.
This allows an improved computational complexity with respect to other algorithms that achieve a similar regret performance.
The authors are clear in the comparison with the main related work.

**Theoretical Claims:**

Yes, with more focus on the proofs of claims relative to the number of resampling iterations.

---

> ### Author Rebuttal · Authors · 2025-03-30
>
> Thank you for your thorough review and valuable feedback on our work. We have addressed your questions and comments below.
>
> **Q1. What is the main technical difficulty in designing a variant of GR that is a sort of counterpart of Tsallis-INF with the Reduced-Variance estimator, especially compared to using importance weighting?**
> **A1.** The main technical difficulties in constructing the Reduced-Variance (RV) estimator for FTPL with GR are as follows:
>
> The RV estimators can take negative value, which makes the analysis significantly difficult. Owing to this issue, after Zimmert \& Seldin (2021) proposed RV estimator, there are still some papers on FTRL that only considers the Importance-Weighted estimator (Jin et al., 2023; Tsuchiya et al., 2023).
>
> Moreover, for FTPL with GR, since the importance weight $w_{t,i}^{-1}$ is replaced with $\widehat{w_{t,i}^{-1}}$, the design and analysis of the RV estimator is more challenging.
>
> Julian Zimmert and Yevgeny Seldin. "Tsallis-INF: an optimal algorithm for stochastic and adversarial bandits.'' JMLR, 2021.
>
> Tiancheng Jin, Junyan Liu, and Haipeng Luo. "Improved best-of-both-worlds guarantees for multi-armed bandits: FTRL with general regularizers and multiple optimal arms.'' NeurIPS, 2023.
>
> Taira Tsuchiya, Shinji Ito, and Junya Honda. "Best-of-Both-Worlds Algorithms for Partial Monitoring.'' ALT, 2023.
>
> **Q2. Did you actually run all the other experiments for FTPL CGR II-B on your own? If so, was there no meaningful difference compared to FTPL CGR II-U?**
> **A2.** We actually ran all the experiments for FTPL CGR II-B, including the pseudo regret for $K=8$ and $K=32$, running time for $K$ varying from $2$ to $128$, and the number of resampling for $K=8$ and $K=32$.
> We confirmed that there is no meaningful difference compared to FTPL CGR II-U. The detailed results can be found at the this link: https://anonymous.4open.science/api/repo/ICML2025_CGRII_Experiments-9807/file/additional_experiments.pdf?v=86d6493c
>
> **Q3. Assumption on the losses in the adversarial setting: we may assume they depend on $(I_1,\ldots, I_{t-1})$.**
> **A3.** Thank you for the comment. While we mentioned the loss may depend on both $\ell$ and $I$, this phrasing was introduced primarily to clarify that we consider an adaptive adversary rather than an oblivious one.
> Still, we understood that as pointed out it is indeed sufficient to consider the loss as a function of $I$'s as in the literature considering an adaptive adversary (Arora et al., 2012; Zimmert \& Seldin, 2021).
> We will modify the writing accordingly.
>
> Raman Arora, Ofer Dekel, and Ambuj Tewari. "Online bandit learning against an adaptive adversary: from regret to policy regret.'' ICML, 2012.
>
> Julian Zimmert and Yevgeny Seldin. "Tsallis-INF: an optimal algorithm for stochastic and adversarial bandits.'' JMLR, 2021.
>
> **Q4.**
> - **Starting from Section 3, $\sigma_i$ is introduced but it seems to also depend on the round $t$; making this dependence explicit (e.g., writing $\sigma_{i,t}$) is probably clearer.**
> - **Theorem 5 seems more of a remark than an actual theorem.**
>
> **A4.** We sincerely appreciate your helpful suggestion. We will modify the notation accordingly to improve clarity.
>
> **Q5. In Section 5, it would be clearer to also specify how the confidence intervals around the curves were chosen (e.g., standard deviation?).**
> **A5.** Thank you for the valuable suggestion. The confidence intervals in the experimental results represent the standard deviation. We will modify the writing to clarify it.

---

### Decision · Program_Chairs · 2025-05-01

**Decision:**

Accept (poster)

**Comment:**

This paper studies a novel loss estimation procedure for FTPL in multi-armed bandits called Conditional Geometric Resampling (CGR).

While FTPL offers no direct advantage in terms of memory or computational complexity for multi-armed bandits, the techniques of this paper likely apply to problem settings where FTPL is computationally more attractive than FTRL. All reviewers agree that this is a solid contribution which is of interest to the community.